# Investigation of the Effects of Surrounding Media on the Distributed Acoustic Sensing of a Helically-Wound Fiber-Optic Cable with Application to the New Afton Deposit, British Columbia

Sepidehalsadat Hendi[1], Mostafa Gorjian[1], Gilles Bellefleur[2], Christopher D. Hawkes[3], Don White[2]

[1]Geological Engineering, Department of Earth, Ocean and Atmospheric Sciences, University of British Columbia, Vancouver, BC, Canada
[2]Geological Survey of Canada, 601 Booth St., Ottawa, ON, Canada
[3]Geological Engineering, Department of Civil, Geological, and Environmental Engineering, University of Saskatchewan, Saskatoon, SK, Canada

*Correspondence to*: Sepidehalsadat. Hendi (shendi@eoas.ubc.ca), Mostafa Gorjian (mgorjian@eoas.ubc.ca)

**Abstract.** Fiber optic sensing technology has recently become popular for oil and gas, mining, geotechnical engineering, and hydrogeology applications. With a successful track record in many applications, distributed acoustic sensing using straight fiber optic cables has become a method of choice for seismic studies. However, distributed acoustic sensing using straight fiber optic cables cannot detect off-axial strain at high incident angles, hence a helically wound cable design was introduced to overcome this limitation. The helically wound cable field data at the New Afton deposit showed that the quality of the data is highly dependent on the incident angle (the angle between the ray and normal vector of the surface) and surrounding media. A 3D finite element model developed using COMSOL Multiphysics quickly and efficiently assessed the effects of various materials surrounding a helically wound cable for simple geometry, for scenarios corresponding to a real deployment of such cable underground at the New Afton mine in British Columbia, Canada. The proposed numerical modeling workflow could be applied to more complicated scenarios (e.g., non-linear material constitutive behaviour and the effects of pore fluids). The results of this paper can be used as a guideline for analyzing the impact of surrounding media and incident angle on the response of helically wound cable, optimizing the installation of helically wound cable in various conditions, and to validate boundary conditions of 3-D numerical model built for analyzing complex scenarios.

## 1 Introduction

Fiber-optic technology has become popular in geophysical, mining, geotechnical, hydrological, environmental, and oil and gas applications. Fiber optic sensing offers many advantages over conventional sensors, including lower price, lower weight, large-area coverage when compared to point sensors, simultaneous sensing over the entire length of the fiber optic cable, low sensitivity of glass fiber to electromagnetic radiation, and the possibility of use in harsh environments (Madjdabadi, 2016). A comprehensive literature review about the different types of fiber optic sensing technology, in terms of application, modulation and scattering of light, and polarization can be found in Madjdabadi (2016), Miah and Potter (2017), and Hartog (2018). In earth-related engineering, three main fiber optic sensing categories are commonly used: distributed temperature sensing (DTS),

distributed strain sensing (DSS), and distributed acoustic sensing (DAS) (Ranjan and McColpin, 2013; Daley et al., 2013; Hornman, 2015).

Distributed Acoustic Sensing is commonly applied to measure elastic waves in seismic applications. In this case, fiber optic cables replace conventional point sensors such as geophones and accelerometers. DAS has been used in vertical seismic profile (VSP) surveys for over 12 years (Daley et al., 2013), is often used to monitor micro-seismicity associated with hydraulic fracturing (Hornman, 2015), and is used in seismology (Jousset et al., 2018; Lindsey et al., 2019; Sladen et al., 2019; Walter et al., 2020). In terms of environmental application, DAS has been recently used in $CO_2$ sequestration projects to characterize storage reservoirs and to map the progression of the $CO_2$ plume within geological formations (Miller et al., 2016; Harris et al., 2016). In mining, DAS-VSP was used to image steeply-dipping ore at the Kylylahti Cu-Au-Zn deposit in Finland and to provide geological information at the New Afton Cu-Au deposit in Canada (Riedel et al., 2018; Bellefleur et al., 2020).

To date, most DAS applications use cables with straight optical fibers deployed in trenches at surface or in boreholes. Fiber optic cables are most sensitive to seismic waves exerting strain in the axial direction (i.e., longitudinal to the fiber) (Mateeva et al., 2014). For plane compressional waves, the sensitivity of a straight fibre-optic cable varies as a function of $\cos^2 \theta$, $\theta$ being the angle between the plane wave direction and the cable (Mateeva *et al.* 2014, Kuvshinov 1996). Thus, straight fiber optic cables are most suitable for specific survey geometries. These include VSP applications, because the reflected compressional seismic waves propagate predominantly in a direction parallel to the fiber axis, and hydraulic fracturing monitoring, because the monitoring wells are proximal to the induced microseismic events hence the waveforms reaching the monitoring well possess sufficient curvature to generate a detectable component in the direction of the fiber axis. Conversely, DAS can hardly detect seismic waves having particle displacements orthogonal to a straight fiber optic cable due to a lack of resultant axial dynamic strain along the fiber (Hornman, 2015; Ning & Sava, 2016; Innanen, 2017; Eaid et al., 2018; Ning & Sava, 2018; Ning, 2019). With a proper acquisition geometry, DAS can measure not only compressional waves but also mode-converted waves, shear-waves, and surface waves when fibre optic cable is at or near the surface.

The broadside sensitivity of DAS can be improved by using helically wound fiber optic cable (HWC) (Den Boer, 2017). HWC consists of fibers wrapped around a mandrel core with a predetermined wrapping angle (α) (Figure 1). The wrapping angle controls the sensitivity of the cable, with lower wrapping angles providing higher sensitivity to broadside seismic waves. Specifically, a HWC is sensitive to both axial and radial strain, the latter being dependent on the material around the cable.

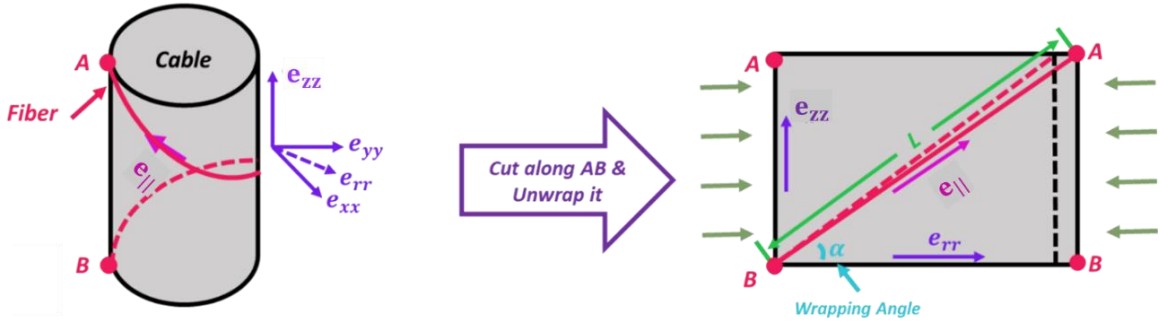

**Figure 1: Illustration showing the meaning of wrapping angle (α). If the cylindrical surface of a cable (grey) is cut along AB, and unwrapped to a horizontal plane, the fiber trajectory would be represented by the diagonal red line shown in the image on the right. The wrapping angle (α) is the angle between the fiber and the circumference of the cable (BB). When a wave hits the HWC, the cable is deformed (dashed black line), and accordingly the fiber is deformed (red dashed line). Dynamic strains (denoted by *e*) imposed on a HWC due to acoustic waves are defined in an x, y, ‖ coordinate system. (After Kuvshinov 2016)**

The properties of the engineered and natural media surrounding the cable and the coupling of these media with the rock formation are important factors controlling the performance of HWC (Kuvshinov, 2016). Here, we present modeling results to help understand the performance of a helically-wound fiber in a field study conducted at the New Afton mine, Canada (Bellefleur et al., 2020). At New Afton, poor coupling and soft formations near the monitoring borehole caused weak seismic amplitudes in data acquired with HWC, but had minimal effects on data measured on a coincident straight fiber optic cable (Figure 2). This experience illustrates the importance of understanding the impact of surrounding media on HWC in selecting and designing DAS monitoring systems.

The main objectives of this work were to help address the following questions:

- Why does HWC sometimes acquire low-quality data in the field?
- What are the effects of the surrounding media and incident angle on their impacts on cable design and installation techniques?

To meet these objectives, we first present a simple adaptation of the 2-D analytical solution of Folds and Loggins (1977) to model HWC's dynamic strain due to acoustic waves. The analytical solution estimates the HWC response using a relatively simple model that does not require specialized software. In the context of this work, the most significant value of this analytical solution is that it provides a means to validate the choice of boundary conditions of a 2-D numerical model developed using the commercial software COMSOL Multiphysics. Then, having established the effectiveness of the 2-D numerical model, a 3-D model was developed using the same methods and tools, by adapting 2D boundary conditions to 3D simulation based on geometrical and symmetry considerations. Material properties and HWC geometry used for the modeling were based on the field conditions and data previously acquired during a survey at the New Afton deposit in Canada. Several scenarios are modelled and compared to help identify near borehole conditions explaining the difference between data acquired with straight and HWC at New Afton, specifically the weak-amplitude data obtained with HWC (Figure 2). The HWC modeling results and

approach used here can provide insights for optimizing field deployments. Service companies could benefit by using this workflow prior to installation of HWC fiber optic cable in the field.

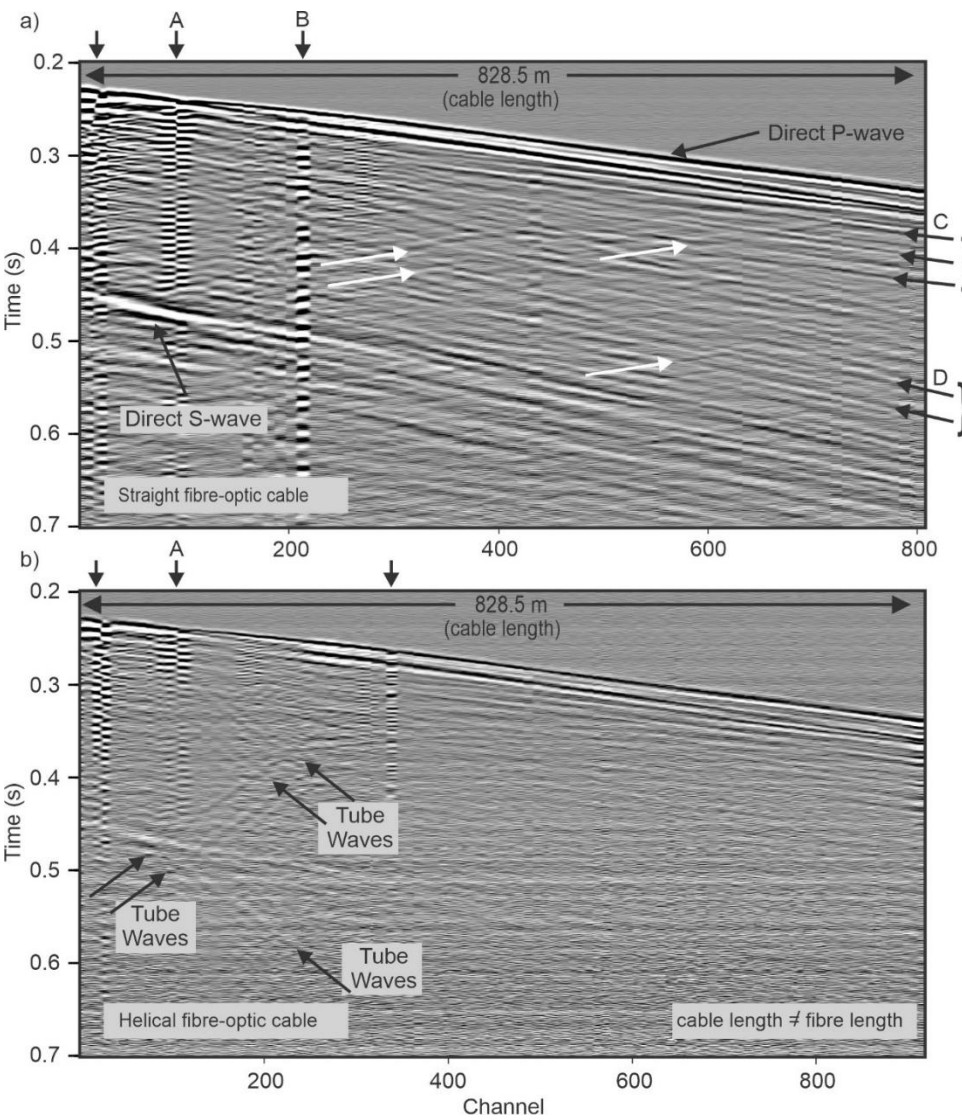

90

**Figure 2. Comparison of field VSP data for a) straight fibre-optic cable and b) helically-wound fibre-optic cable. The helically-wound optical fibre has more channels than the straight fibre for the same length of cable due to the wrapping around the cable core (i.e., 925 channels for helically-wound cable versus 813 for the straight fibre-optic cable). Vertical arrows point to noisy channels indicating poor coupling. White arrows indicate up-going reflections. Arrows C and D point to events of the down-going wavefield**

95 **with moveout of P-waves and S-waves, respectively. Events D arrive before the direct S-wave and are likely the result of P-to-S conversion at a lithological contact or fault zone. The same display gain was used for a) and b). Horizontal banding observed across all channels is optical noise.** Incident angles of seismic waves for data shown in this figure range between 25 and 40 degrees. Grayscale colors show relative seismic amplitudes from negative (white) to positive (black). **Modified from Bellefleur et al (2020).**

## 2 DAS data at the New Afton mine

The modeling work presented in this paper is used to help understand the performance of a helically-wound fiber from a VSP field study at the New Afton mine (Bellefleur et al., 2020). Straight and helically-wound fibre optic cables were deployed in a single borehole to assess the efficiency of DAS to detect geological interfaces and structures associated with mineralization. The New Afton deposit is a porphyry deposit comprising primarily disseminated Cu-Au mineralization. The fibre-optic cables were deployed in a steeply-dipping (70° from horizontal) deviated borehole starting in a work bay located 650 m below the surface and ending at a depth of approximately 1300 m. Both cables were placed inside steel drill rods (used as casing) and cemented in place with grout. The grout was circulated to the bottom of the borehole via a grout tube located inside the casing until grout eventually reached surface from both inside and outside of the casing (drill rods). The grout cured for one month prior to the VSP survey. Based on the afore-noted grout returns both within and outside the casing, it was assumed at the time that grout had filled both the casing and the casing-formation annulus; however, data collected during the survey (discussed below) suggests this may not have been the case. The data were acquired with 1 kg of explosives fired in a 20 m deep shot hole at surface.

An advanced DAS system providing higher signal-to-noise ratio (Carina system by Silixa) was used to record the seismic data on both straight and HWC. Gauge length was 10 m for both straight and helically wound DAS data. Due to the wrapping, strain rate over the gauge length is measured over a shorter cable distance for a helically-wound fibre than for a straight optic-fibre. This resulted in more channels for the HWC data (925 for HWC vs 813 for straight fibre-optic) for cables with identical length (828.5 m). More details about instruments and acquisition parameters can be found in Bellefleur et al. (2020).

Figure 2 presents a comparison of raw field VSP data acquired on collocated straight and helically wound fiber-optic cables at the New Afton mine. The seismic data (or strain rate) for the straight cable (Figure 2a) shows typical downgoing P- and S-waves (black arrows) and some reflected waves (white arrows). Surface waves were likely generated at the source but they are not observed on the DAS data which start at a depth of 650 m below the surface. Also note that almost all reflections (Figure 2a) have a P-wave moveout velocity and are standard P-wave reflections. In comparison, the seismic data obtained with the HWC is characterized by significantly weaker downgoing waves and the lack of reflections (Figure 2b). Tube waves are also present on the HWC data but not on data measured with the straight fibre-optic cable. The modeling work presented below aims at evaluating the effect of the surrounding media on HWC data and assessing if it could explain the overall low data quality in figure 2b.

# 3 Methodology

## 3.1 2D Analytical Modeling

As presented in detail in Appendix A, the wave equation was solved to determine dynamic strain as a function of incident angle generated by dynamic stresses imposed by plane seismic waves propagating through a multilayered medium (after Folds and Loggins, 1977). Plane waves were assumed to represent a point source located in the far-field. Longitudinal-wave properties of the materials were included in the derivation. Equations derived for this model are presented in Appendix A in a form which facilitates comparison with the two-dimensional numerical simulation described later, specifically to confirm the boundary conditions and assumptions applied in numerical simulation. Validation and verification of the numerical model with analyticl results is a standard practice in finite-element modeling (Brinkgreve & Engin, 2013). The pseudocode of our algorithm and procedure for the implementation used in this paper are presented in appendix B. The analytical method in Appendix A is simpler to implement than the analytical approach of Kushinov (2016) especially when a large number of layers surround the fiber-optic cable. In terms of considering geometry in calculations, Kuvshinov's technique is more flexible. However, the analytical method proposed by Kuvshinov (2016) is more accurate as it evaluates cable strain in a layered cylindrical geometry. In this paper, results obtained with the simpler approach is sufficient to support and validate the choice of boundary conditions used for 2D finite element modeling (e.g. see comparison of analytical and finite element results presented later in the paper). Specifically, 2D analytical and 2D finite element-modeling methods use the same geometrical configuration (i.e. multilayered 2D medium). Thus, the more accurate method of Kuvshinov is not appropriate for comparison with 2D finite element modeling results. Once established in 2D, boundary conditions can be determined for 3D finite-element modelling based on geometrical and symmetry considerations.

## 3.2 Finite Element Numerical Model

The numerical models for this work are based on the finite element method and were developed using the COMSOL Multiphysics software. The model predicts the axial strain of a fiber as a function of the incident angle of a planar compressional wave. S-wave could also be modeled for isotropic cases by considering that particle displacement is orthogonal to the wave propagation direction and adjusting the incidence angle accordingly. Finite element modeling can handle the complexity of realistic geological conditions and is not subject to the simpler geometric assumptions used in analytical modeling (i.e., for instance planar or cylindrical assumptions). We simulated only incident P-waves because it is the primary body wave used for exploration at the New Afton mine. As shown in Figure 2, the reflected wavefield at New Afton is dominated by P-wave reflections, whereas S-wave and mode-converted wave reflections were not identified with high certainty.

The models were developed in the frequency domain, using a dominant frequency of 100 Hz (wavelength of approximately 40 m)  in order to match the field data at New Afton. Since the maximum response of a fiber as a function of incident angle was the main objective, simulations were conducted in the frequency domain. Dominant frequency is deemed to have an effect on the transmitted energy to the fiber in an absolute sense, but not on the relative magnitudes of the results generated for a range of scenarios. We do not expect significant variations in the incident angle yielding maximum response over the range of frequencies typical of seismic surveys for mineral exploration (5-150 Hz).

Six scenarios were modeled in the frequency domain with different mechanical properties for the materials within and around the borehole. These scenarios are listed in Table 1, and were chosen to enable assessment of the effects of various potential geometries, materials and conditions on DAS data measured with a HWC. For example, scenarios #1 and #2 compare the effect of soft and hard cement between casing and rock formation, whereas scenarios #3 and #4 compare the effect of soft versus hard formations. Scenarios #5 and #6 investigate the effect of water located either inside or outside the casing. For this work, the geometries (Table 2) and properties (Table 3) of the materials were chosen to be similar to those of a DAS field experiment conducted at the New Afton mine, Canada (Bellefleur et al., 2020). The rock properties were based on unpublished triaxial compression testing results obtained for representative samples of crystalline fragmental volcanic host rocks at the New Afton deposit. Samples were retrieved at depths ranging from 2 to 214 m. The average values of elastic properties are presented in Table 3. The material properties related to the cable, casing and hard cement were taken from Kuvshinov (2016). The six scenarios are modeled. The number and type of layers in order of proximity to the center of the cable (domains) are given in Table 1. Inside the domain, a tetrahedral mesh was employed, while lower reflecting surfaces were covered with a swept mesh (see figure 4).

**Table 1. Number and type of layers (domain) implemented in numerical modeling for six scenarios. For the 2-D numerical modelling, the layers are planar. For 3-D numerical modelling, the layers are concentric, as illustrated in Figure 3.**

| Scenario # | Number of layers | Type of layers (in order of proximity to the center of the cable) | Figure |
|---|---|---|---|
| 1 | 5 | cable, hard cement, casing, hard cement and hard formation | 3a |
| 2 | 5 | cable, hard cement, casing, soft cement and hard formation | 3a |
| 3 | 4 | cable, hard cement, casing and hard formation | 3b |
| 4 | 4 | cable, hard cement, casing and soft formation (soft cement) | 3b |
| 5 | 5 | cable,  cable, hard cement, casing, water and hard formation | 3c |
| 6 | 5 | cable, water, casing, hard cement, and hard formation | 3d |

185

. **Table 2. Geometrical parameters used in numerical modeling.**

| Material | Geometrical parameter | Value |
|---|---|---|
| Cable | Cable Diameter | 25.0 mm |
| | Cable Height | $\eta/12$ [1] |
| Cement/water (around the Cable) | Cement Diameter | 77.8 mm |
| | Cement Height | $\eta/12$ |
| Casing | Casing Diameter | 88.9 mm |
| | Casing Height | $\eta/12$ |
| Cement/water (around the Casing) | Cement Diameter | 96.0 mm |
| | Cement Height | $\eta/12$ |
| Formation | Formation Width* | $\eta/12$ |
| | Formation Height* | $\eta/12$ |

* Size of the model for hard and soft formations are $4.78^3$ and $1.63^3$ m$^3$

**Table 3. Layer (domain) properties used for modeling**

| Material | $\rho$ (kg m$^{-3}$) | Compressional Velocity ($V_p$) (m s$^{-1}$) | E' [2] (GPa) | $\nu$ |
|---|---|---|---|---|
| Hard Formation | 2734 | 5736 | 60.1 | 0.28 |
| Cable | 1200 | 1183 | 1.6 | 0.15 |
| Casing | 8050 | 5635 | 200.0 | 0.28 |
| Hard Cement | 2240 | 2728 | 15.0 | 0.20 |
| Water | 1000 | 1500 | N/A | N/A |
| Soft Cement/Soft Formation | 1440 | 1963 | 5.0 | 0.20 |

190

---

[1] $\eta = \dfrac{(Formation\ Compressional\ Velocity)}{(Dominant\ Frequency)}$

[2] E' is plane strain Young's modulus which is equivalent to $E/(1-\nu^2)$, where E is Young's modulus and $\nu$ is Poisson's ratio

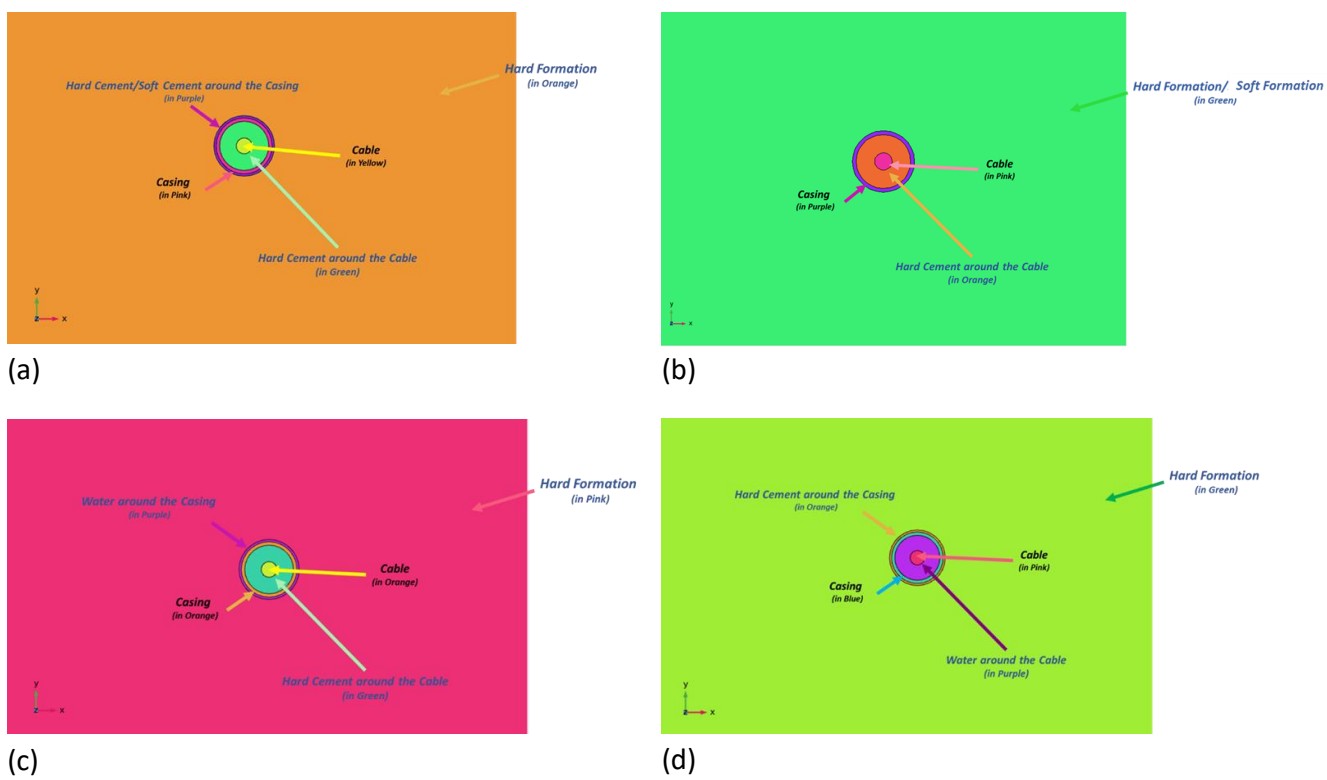

**Fig. 3. 3-D model geometries for various scenarios, as follows: (a) #1 and #2, (b) #3 and #4, (c) #5, (d) #6**

In all scenarios, strain was simulated for a helically wound cable with a wrapping angle (α) of 30°(This wrapping angle is derived from the HWC cable used in the New Afton mine; see Figure 2). Similar to the analytical results and as shown in Appendix A (equation A.49), the longitudinal and radial strains of the cable were modelled separately and combined together with the wrapping angle to obtain the resultant axial strain of the fiber. As mentioned previously, the impact of the surrounding media is best observed on the radial strain. Thus, modelling strains separately allows to show details of radial strain and assess the effect of surrounding media more effectively.

### 3.3 Boundary Condition for Numerical Modeling

The geometry and boundary conditions used for the 2-D numerical simulations are shown in Figure 4a. This model treats the material domains around the borehole as planar, rather than concentric. A prescribed displacement boundary condition was selected for the right edge of the model in order to model arrival of the incoming wave on this surface (i.e., by prescribing dynamic strains representing a compressional wave). A low reflecting boundary was selected on the left because it is assumed there is no reflection from the adjacent semi-infinite domain. Periodic boundaries were chosen on the top and bottom boundaries to make the solutions equal on both sides.

The geometry and boundary conditions used for the 3-D simulations are presented in Figure 4b. They are the same as the 2-D simulation, except the material domains are concentric, and centered around the cable. The low reflecting boundary condition is applied to the other boundaries as depicted in Figure 4.b. Inside the domain, a tetrahedral mesh was employed, while lower reflecting surfaces were covered with a swept mesh (Figure 5).

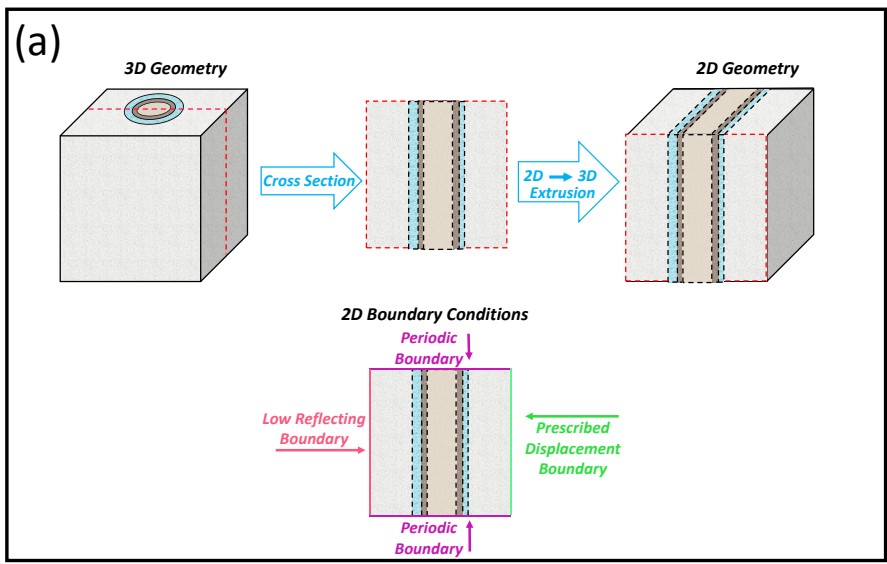

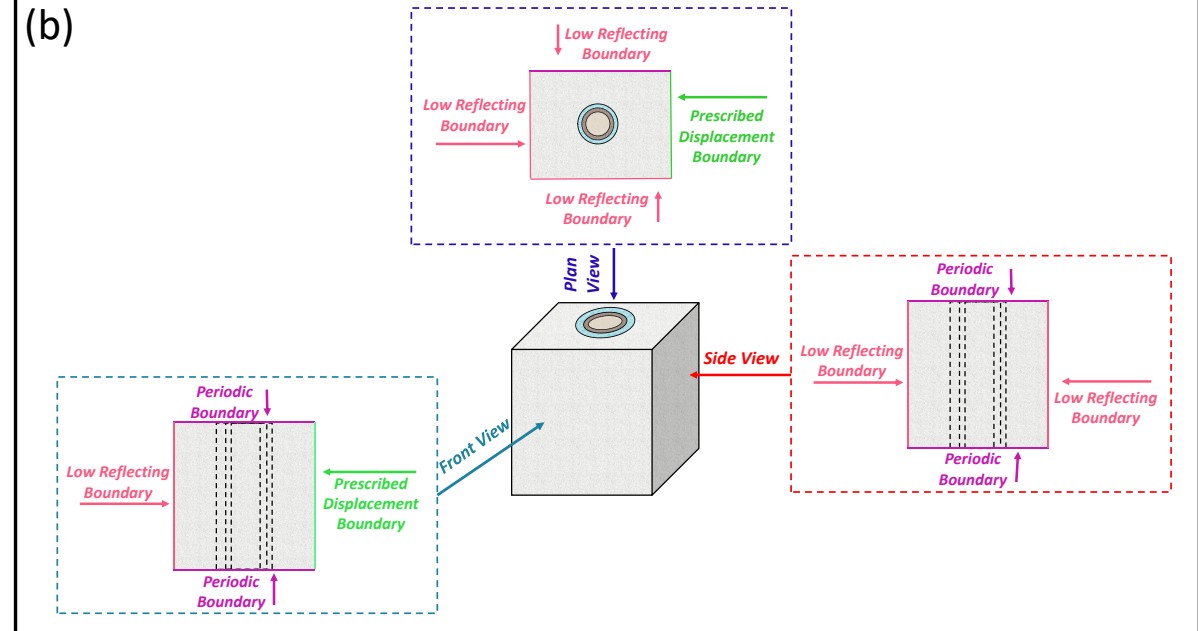

**Figure 4. (a) Illustration showing how a cross-section of the actual 3-D geometry was used to create a plane-strain, 2-D numerical model, and the boundary conditions applied to this model, (b) Assigned boundary conditions for the 3-D model.**

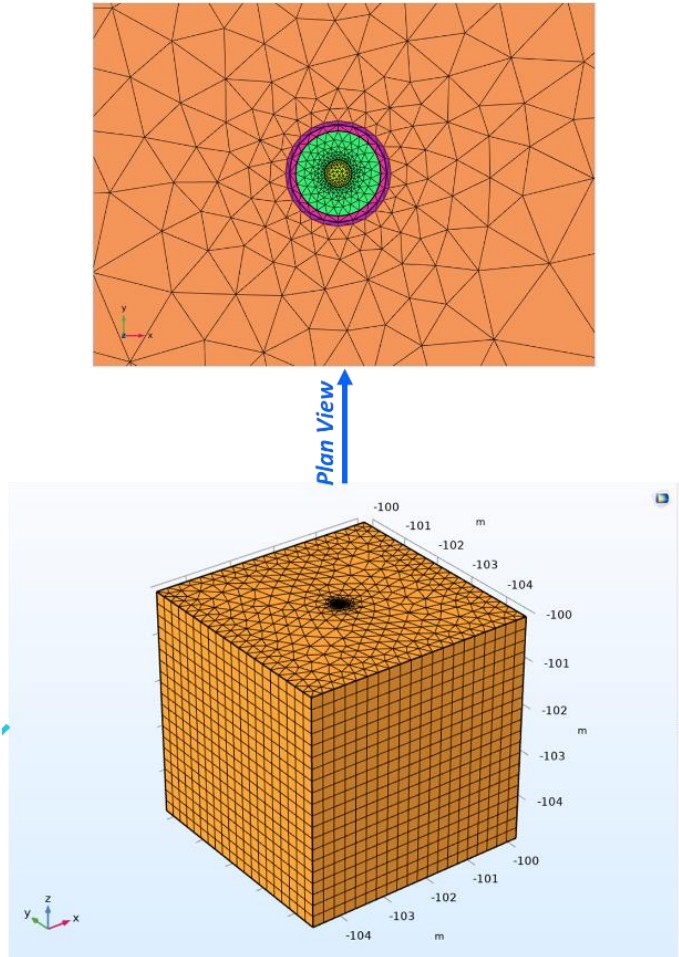

**Figure 5. Mesh generation with COMSOL. Inside the domain, a tetrahedral mesh was employed, while lower reflecting surfaces were covered with a swept mesh.**

## 4 Results

### 4.1 Model Validation and 2D Sensitivity Analysis

Scenarios #1 to #4 were modeled using both 2D analytical and numerical models in order to: (i) validate the choice of boundary condition in numerical model, and (ii) to investigate the sensitivity of fiber response to cement stiffness and formation stiffness.

The numerical and analytical results for scenarios #1 to #4 are presented in Figures 6 to 7. In all cases, the agreement between

numerical and analytical results is good, with root mean square errors (calculated for the difference between the analytical and numerical solutions) in the 0.003 to 0.004 range. The effect of the stiffness of the cement between the rock formation and casing is shown in Figure 6. The analytical responses are almost identical for both scenarios whereas numerical responses differ slightly for low angles of incidence. At those angles (0-35°), fiber strain is less when hard cement surrounds the casing.

The effect of the stiffness of cement around the casing becomes negligible for high angles of incidence. The effect of rock formation stiffness on fiber strain is particularly significant (Figure 7). The fiber strain is lower for a soft formation, modeled here using the properties of soft cement, than for a hard rock formation. This should result in a weaker seismic signal measured with DAS for soft rock formations. Scenario 4 differs from the others in that the soft formation is adjacent to the casing in this case. Because of the huge difference in material properties, the reflection index on this surface is larger, and a large quantity

of energy is reflected to the soft formation. As a result, the maximum strain in this situation is smaller than in others

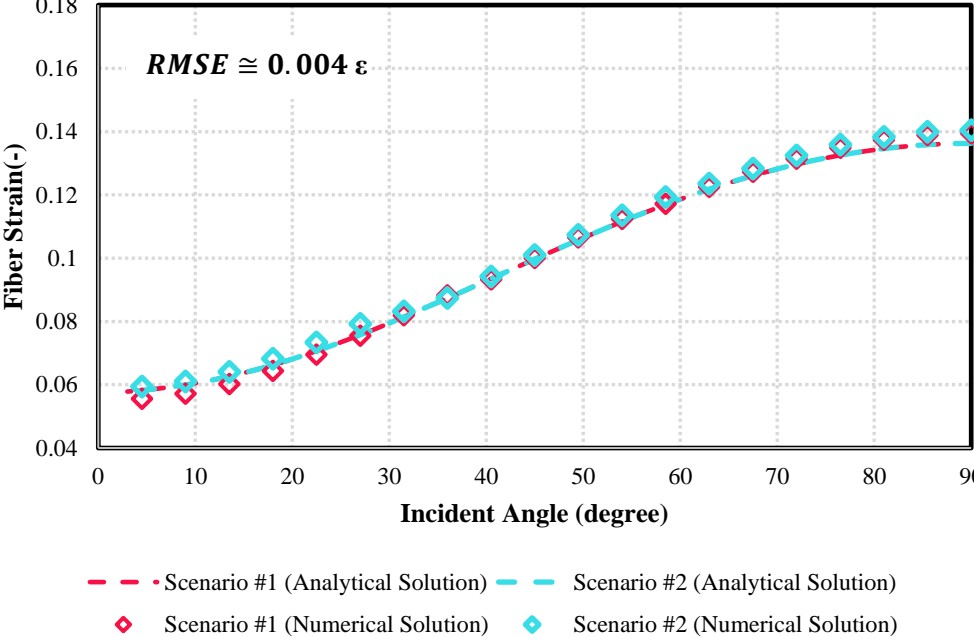

**Figure 6. Axial strain for HWC fiber predicted using two-dimensional numerical model and analytical solution for scenarios #1** (cable, hard cement, casing, hard cement and hard formation) **and 2** (cable, hard cement, casing, soft cement and hard formation)**. RMSE denotes the root mean square error calculated for the difference between the analytical and numerical solutions.**

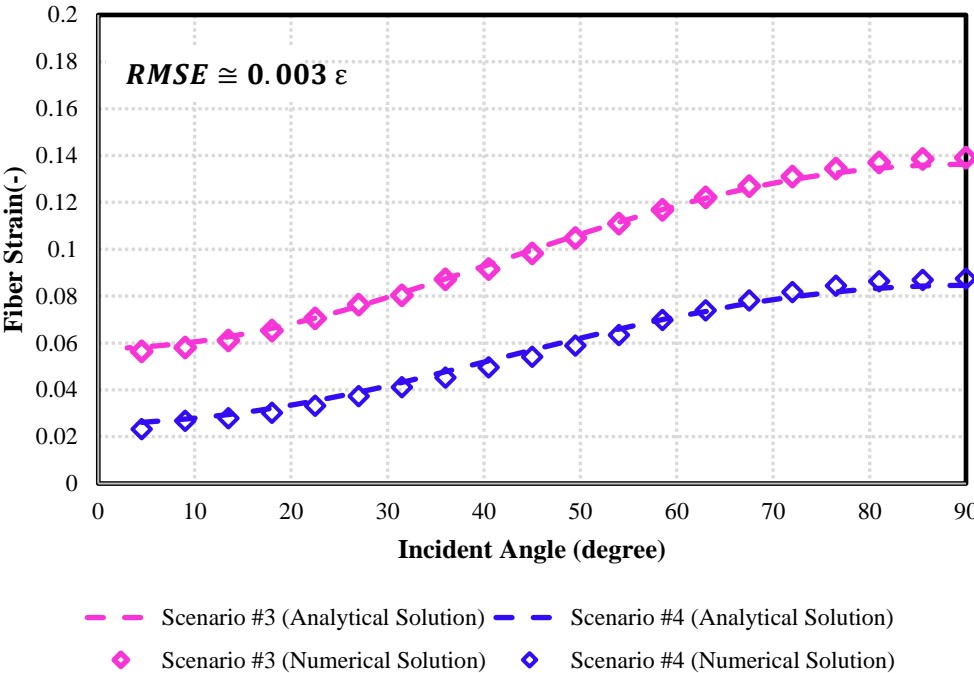

**Figure 7. Axial strain for HWC predicted using two-dimensional numerical model and analytical solution for scenarios #3** (cable, hard cement, casing and hard formation) **and #4** (cable, hard cement, casing and soft formation). **RMSE denotes the root mean square error calculated for the difference between the analytical and numerical solutions.**

### 4.2 3D Numerical Modeling Results

Radial strain distributions generated using the three-dimensional numerical model for scenarios 1 to 6 are shown in Figures 8-13. Those figures show the strain in all layers for a horizontal section of the 3D model, which cannot be obtained easily with analytical methods. Axial strain results were also obtained from COMSOL and used to calculate helically wound cable strain $(e_{zz_{(Fiber)}})$ by considering a wrapping angle $(\alpha)$ equal to 30 degrees, according to equation A.49 (Appendix A). Scenarios with 5 layers and several thin layers (scenarios #1, #2, #5, and #6) required a significantly finer mesh in the region between the formation-casing annulus and the cable. Though valid results were obtained, graphical rendering of strains in these finely meshed domains caused some issues, particularly for strain inside the cable. Rendered strains inside the cable for scenarios #1, #2, #5, and #6 are not displayed correctly in Figures 8, 9, 12, and 13, respectively. Strains inside the cable for scenarios #3 and #4 (Figs. 9 and 10) are correctly displayed. Despite any graphical rendering issues, valid numerical strain values were obtained for all elements in all scenarios, and these results were used to generate graphs of fiber strain versus incident angle, as shown in Figures 14 through 17.

Figures 8 and 9 compare the radial strain in cases where hard or soft cement fills the casing-formation annulus, respectively. Strain in the formation for both scenarios are similar and have maximum values along the x-axis, which corresponds to the direction of plane wave propagation. The main difference is the higher strain observed in the thin layer of soft cement between the casing-formation annulus. Strain in the hard cement between cable and casing is, however, similar. . The effect of having a hard or soft rock formation on radial strain is shown in Figures 10 and 11, respectively. The soft formation has higher strain than the hard formation. The maximum value of strain for the soft formation, contrary to the hard formation, is distributed along the y-axis (Figure 11). This distribution of maximum strain along the y-axis is also observed inside the cable (Figure 11). The difference is pronounced when comparing with strain inside the cable for the hard formation (Figure 10). Also note the significantly higher radial strain inside the cable when the borehole is within a hard formation. Finally, figures 12 and 13 compare the effect of having water inside the casing or outside the casing between the rock formation. The distribution of strain in the formation in the immediate vicinity of the borehole is significantly different for both scenarios with four areas of maximum strain occurring when water fills the inside of the casing. Strain in the casing-formation annulus is however larger with filled with water. For the reason mentioned above, strain inside the casing and inward cannot be compared on those figures. However, a comparison of strain as a function of angle of incidence for these scenarios is shown in Figure 16.

Figure 14 shows a comparison of modeled fiber strains for scenarios #1 and #2. These are identical 5-layer scenarios, except #1 assumes hard cement in the casing-formation annulus, whereas #2 assumes a soft cement in this annulus. The results (fibre strain) are nearly identical for incident angles less than 30°. However, for larger incident angles, the scenario with soft cement in the annulus (i.e., scenario #2) is slightly more sensitive (i.e., nearly 10% greater fiber strain). This is interpreted to be due to the lower Young's modulus for soft cement, resulting in the fiber experiencing greater strain in response to an imposed dynamic stress. It should be noted that the analytical results for scenarios #1 and #2 (shown in Figure 6), showed a smaller difference between cable strain at high incident angles. A possible reason for this is that in two-dimensional analytical analysis, strain in the y-direction was considered zero under the plane strain assumption. In 3D, the cylindrical casing would have a slight dampening (shielding) effect on strains transmitted to the inner cement and fiber cable, but in 2D a planar "casing" would have no such dampening effect. However, results between 2D and 3D simulations cannot be directly compared quantitatively. This is explained by the fact that energy of the source plane wave and deformation occur only in the x-z plane. The strain values from 3D finite element simulations are typically less than those obtained from 2D simulations.

Figure 15 shows a comparison of modeled fiber strains for scenarios #3 and #4. Both of these are 4-layer scenarios which assume that casing is in direct contact with the formation; as such, there is no casing-formation annulus. The difference between these scenarios is the fact that #3 assumes a hard formation, whereas #4 assumes a soft formation. The results are nearly identical for incident angles less than 5°, but with increasing angle scenario #3 shows a much stronger response (strains at 90° angle nearly five times greater to #3 versus #4). For the same frequency (100 Hz), the wave number of the hard formation is more than that of the soft formation. As a result, amplitude decrease is higher in the soft formation than in the hard formation.

Accordingly, less seismic energy is transmitted to the cable in scenario #4, compared to scenario #3, resulting is less strain for scenario #4.

Figure 16 shows a comparison of modeled fiber strains for scenarios #5 and #6. These are both 5-layer scenarios similar to scenario #1, except for the presence of water either between the casing-formation annulus (scenario #5) or between the cable-casing annulus (scenario #6). Hard cement replaces water when not present for both scenarios. Scenario #6 shows tensile strains at low incident angles (less than $18°$). However, the increase in strain with incident angle is relatively high compared to scenario #5, hence the compressional fiber strains for scenario #6 exceed those for scenario #5 for incident angle greater

than 25°. Overall, the results show that the existence of water in the media surrounding the cable results in lower fiber strains than most of the other scenarios. More specifically, the maximum strain (which occurs at 90° incident angle for all scenarios) is roughly 0.0065 for scenario #5 and 0.0185 for scenario #6, compared to values in the 0.027 to 0.05 range for scenarios #1 to #3. (The only other scenario with a comparably low strain, at 0.009, is scenario #4 which, as discussed in the previous paragraph, assumes a soft formation). The low strains observed for scenarios #5 and #6 are mainly due to the interfaces between

solid and water forming a free surface, which decompose incident waves into interface waves (i.e., tube waves) and compressional waves. Accordingly, less compressional energy is transmitted to the fiber cable. Though this is true of both scenarios #5 and #6, strains are greater in #6 because cable extension/contraction is uninhibited when the cable is within water rather than cement.

Figure 17 shows a comparison of modeled fiber strains for scenarios #1 and #3. These are based on the same material

properties, except scenario #1 includes a cement-filled casing-formation annulus (5-layer system) whereas scenario #3 assumes direct contact between the casing and formation (4-layer system). Both scenarios show relatively high fiber strains, with scenario #3 having the most significant strains of all scenarios considered; and scenario #1 having the third-highest (only slightly less than scenario #2). As such, the number of layers, as a stand-alone factor, should not be expected to result in poor signal quality for a HWC, assuming the layers present are hard (i.e., stiff) and well coupled.


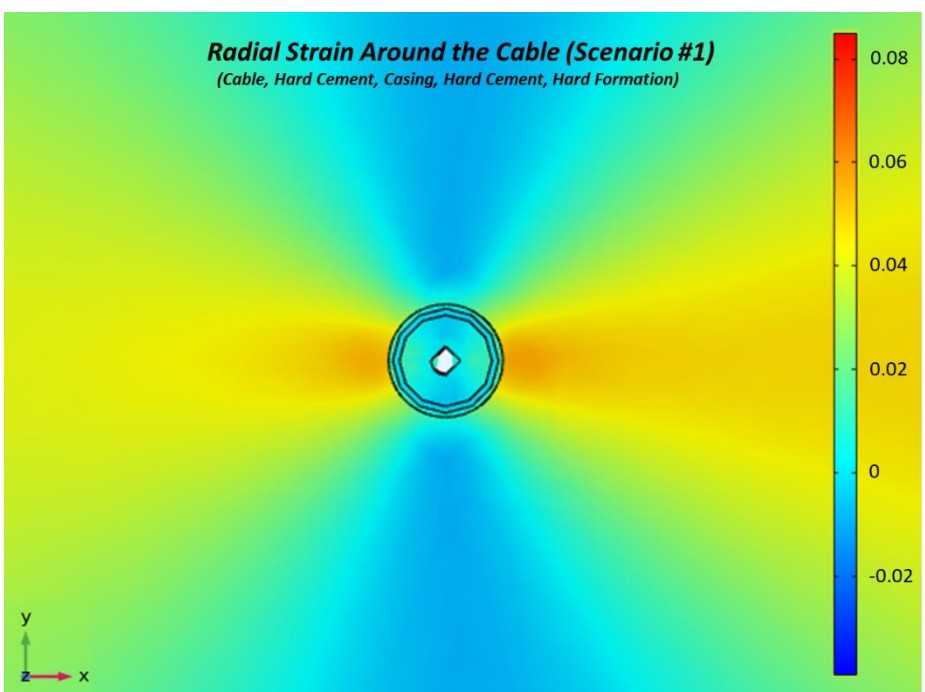

**Figure 8. Radial strain around the Cable as a response of compressional wave propagation in 90° incident angle, Scenario #1 (incident waveform is incoming from the right).**

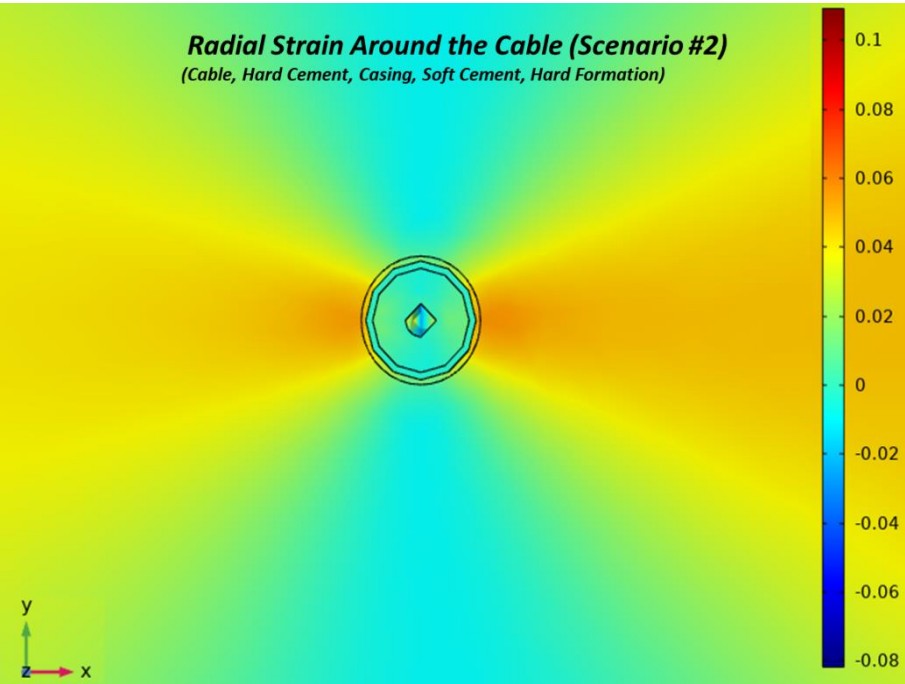

**Figure 9.  Radial strain around the Cable as a response of compressional wave propagation in 90° incident angle, Scenario #2 (incident waveform is incoming from the right).**

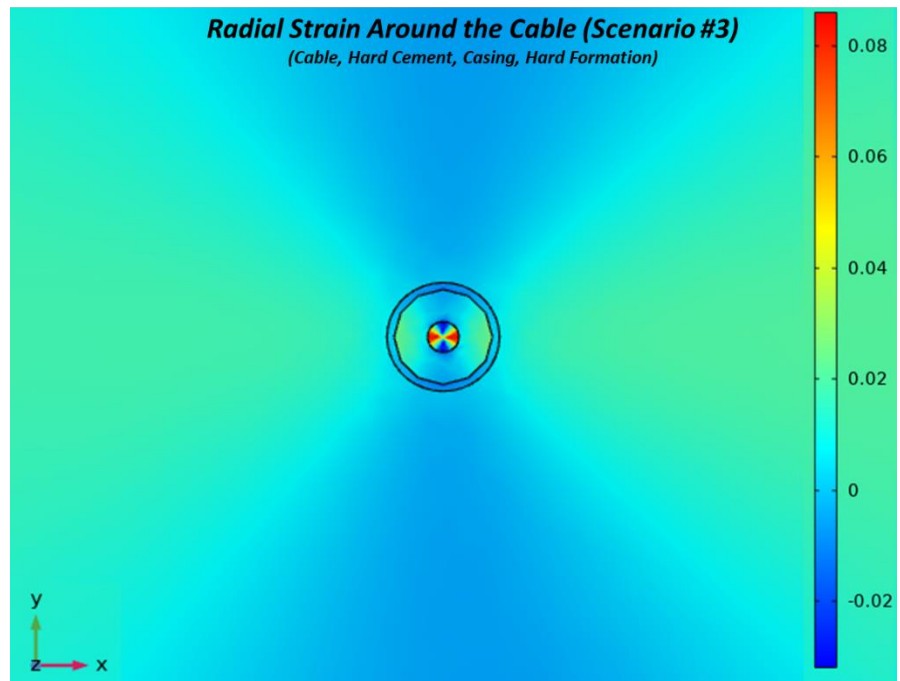

**Figure 10. Radial strain around the Cable as a response of compressional wave propagation in 90° incident angle, Scenario #3 (incident waveform is incoming from the right).**

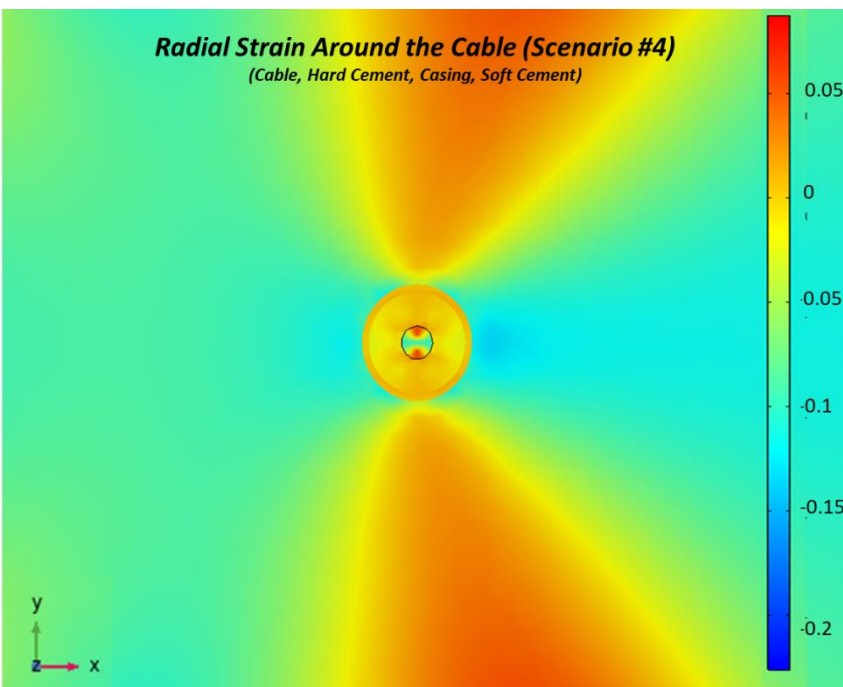


**Figure 11. Radial strain around the Cable as a response of compressional wave propagation in 90° incident angle, Scenario #4 (incident waveform is incoming from the right).**

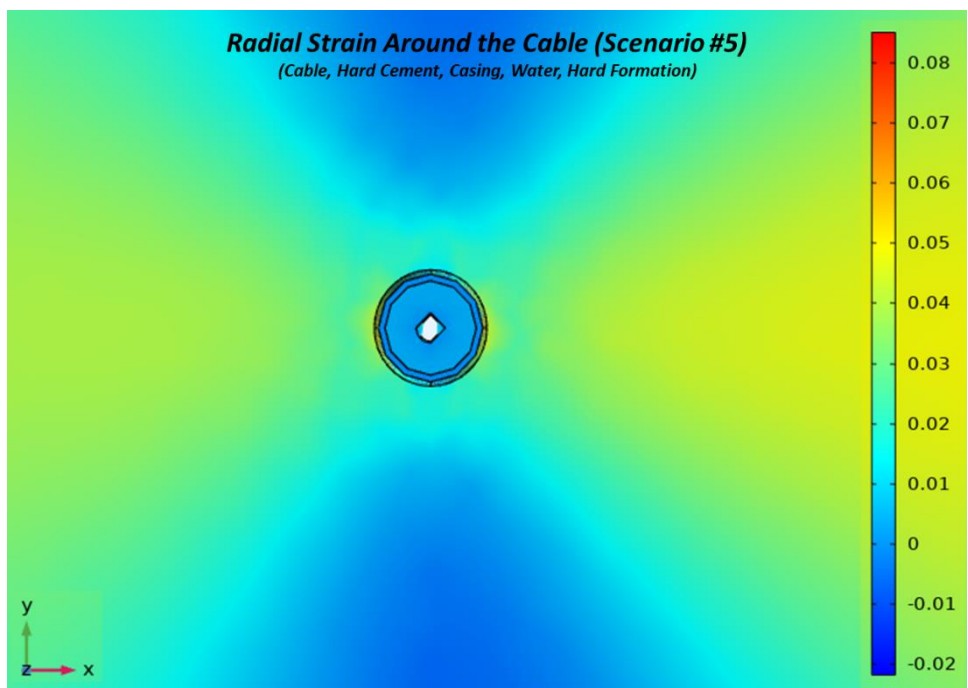

**Figure 12. Radial strain around the Cable as a response of compressional wave propagation in 90° incident angle, Scenario #5 (incident waveform is incoming from the right).**

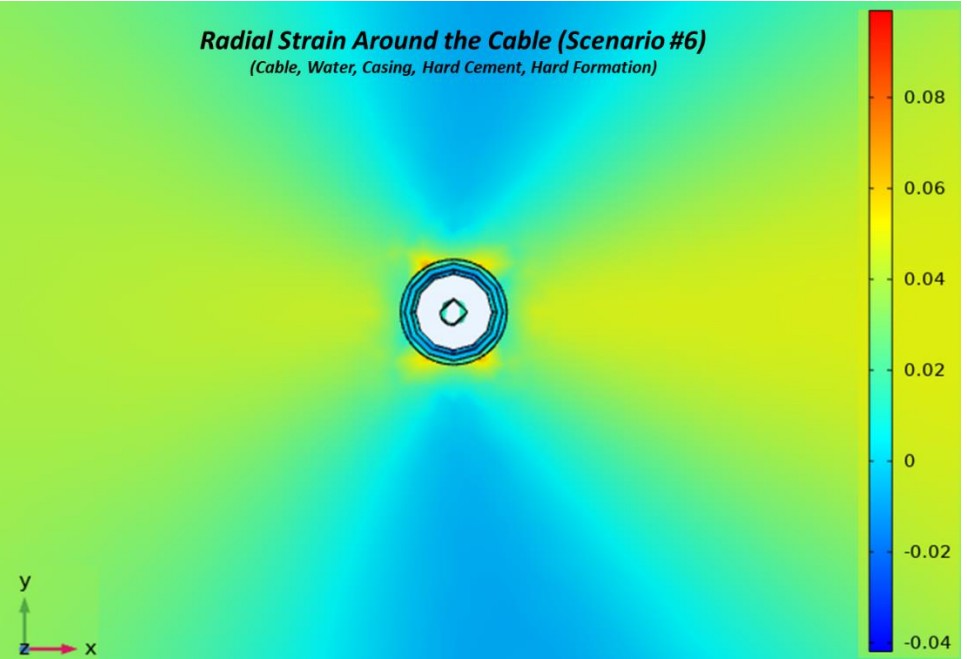

**Figure 13. Radial strain around the Cable as a response of compressional wave propagation in 90° incident angle, Scenario #6 (incident waveform is incoming from the right).**

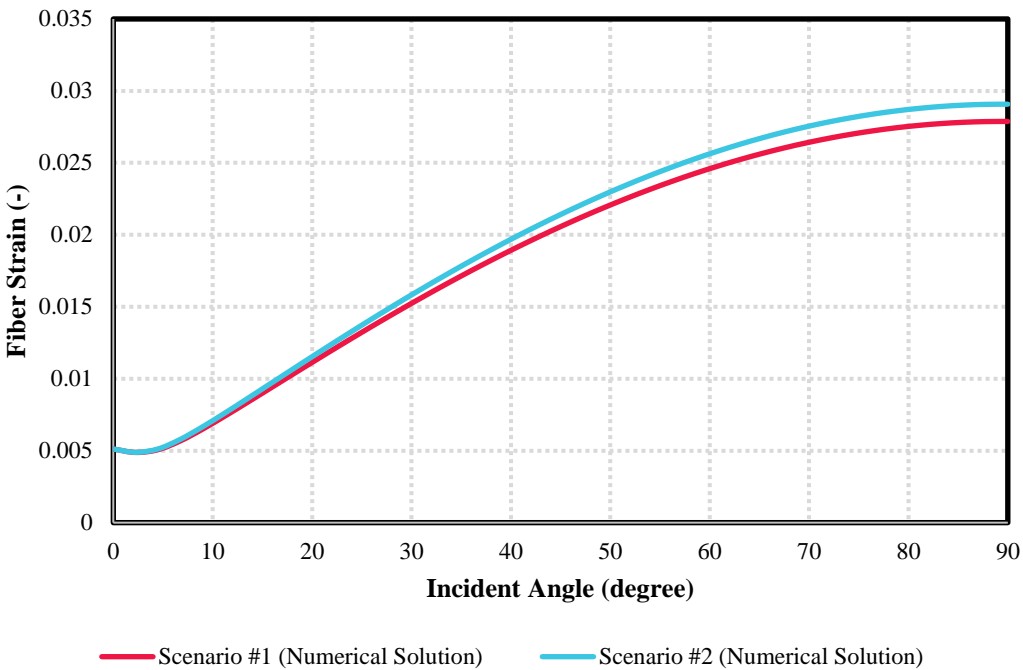

Scenario #1 (Numerical Solution)    Scenario #2 (Numerical Solution)

**Figure 14. Investigation the effect of cement quality (soft vs. hard) surrounding the casing on the acoustic response of fiber (comparison between scenarios #1 and #2).**

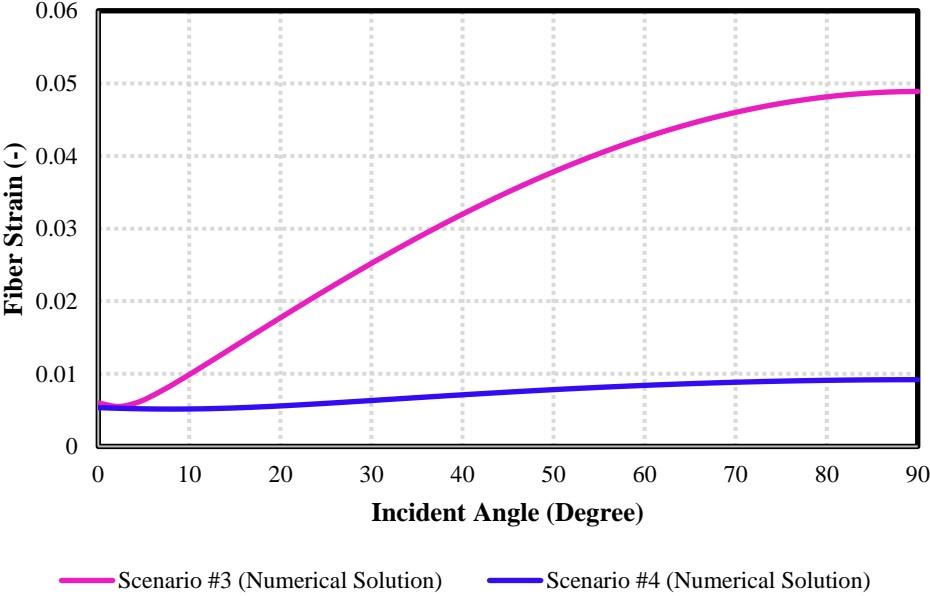

Scenario #3 (Numerical Solution)    Scenario #4 (Numerical Solution)

**Figure 15. Investigation the effect of hard/soft formation adjacent to the casing on the acoustic response of fiber (comparison between scenarios #3 and #4).**

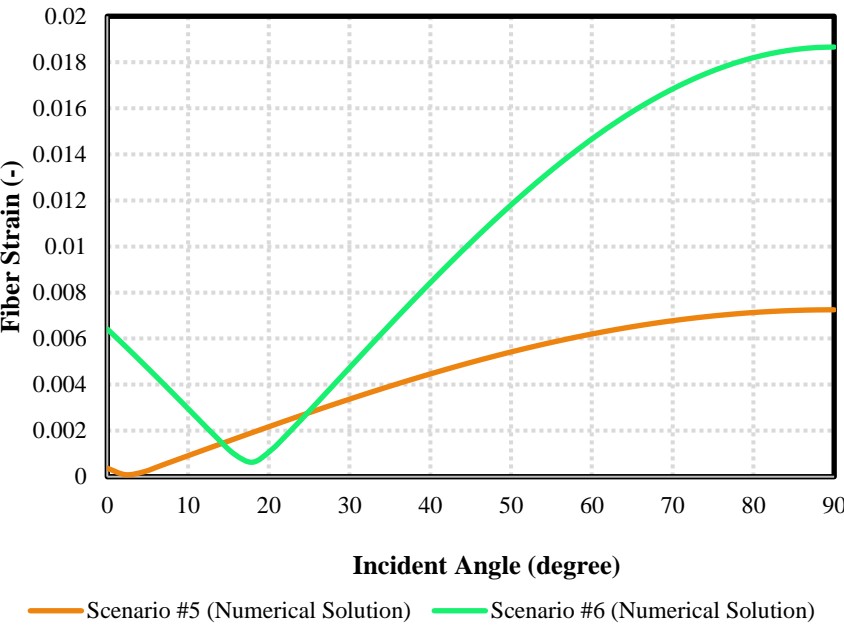

**Figure 16. Investigation the effect of water on the acoustic response of fiber (comparison between scenarios #5 and #6). The strain are absolute values, and the strains for incident angle below 18° are tensile.**

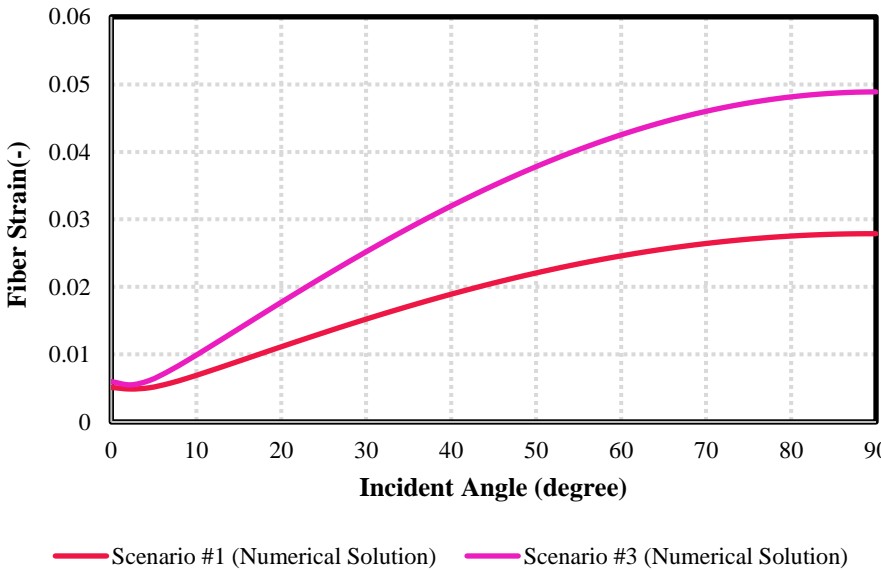

**Figure 17. Investigation the number of layers on the acoustic response of fiber (comparison between scenarios #1 and #3).**

**5 Discussion**

Bellefleur et al. (2020) make the following observations for the New Afton DAS data and offer the following interpretations to explain them. First, data recorded with the straight fiber-optic cable include several noisy traces at the locations indicated by the vertical arrows near the top-left corner of Figure 2a; in particular near traces 90 and 215, which coincide with fault zones as identified using wireline log data. Such noise is typical of un-cemented or poorly cemented casing and is caused by local casing resonance. Second, Noisy traces are also observed at similar locations in the HWC (see vertical arrows near the top-left corner of Figure 2b), but resonance noise on those traces is not as strong as the straight fiber data. The HWC data are, however, strongly affected by tube waves between the two fault zones (see diagonal arrows in the left half of Figure 2b), which suggests the presence of liquid or incomplete cementing of the casing-formation annulus. Finally, aside from the traces showing noise and resonance, the signal strength (fiber strain) of the linear fibre appears relatively good. As shown in Figure 2a, this dataset contains many events indicated with arrows and comprising direct P- and S-waves, many down-going waves (arrows C and D), and reflected waves (indicated with white arrows). This suggests that the cement within the casing (i.e., encompassing the fibre cables) cured properly and contained no liquid, hence enabling a strong signal in the axial direction in response to the vertical component of the propagating seismic waves. Conversely, the HWC dataset showed a relatively weak response to the afore-noted events.

While the modeling scenarios considered in this work do not fully capture the complexity of ground conditions and coupling behind the casing (inferred from DAS data), they offer some explanation for some of the observations made above. Based on the results obtained in this work, the presence of a soft formation (scenario 4) or water (scenario 5) outside the casing are likely to reduce the amplitude measured on a HWC located inside the casing. At New Afton, faults zones consist of weakly-competent and brittle rocks that could likely be considered similar or even weaker to the soft formation used in our models. The presence of water outside the casing is also indicated locally by tube waves on the HWC data. For scenario #5 (water-filled casing-formation annulus), the relatively weak response of the HWC dataset suggests that cementing emplacement and curing outside of casing was not effective for a significant portion of this borehole. A casing properly coupled with rock formation with either soft or hard cement (scenarios 1 and 2) would not impede HWC measurements. The results obtained in scenario #4 (soft rock formation in direct contact with casing) provide an additional explanation for poor signal quality; one which might be relevant over some intervals of the borehole. This is suggested because Bellefleur et al. (2020) noted several weak and unstable intervals (including fault zones), and had to ream these intervals during drilling and logging operations, and leave drill rods in place over these intervals to prevent borehole collapse during logging operations. Scenario #4 suggests that, even if the weak zone converged on the casing and achieved reasonably effective coupling, the soft nature of the rocks in these intervals would result in low fiber strains for the HWC.

The DAS dataset at New Afton, interpreted in the context of our modeling, serves as a practical demonstration of the extreme effects of surrounding media and coupling on HWC data quality. These results, and new scenarios simulated with the models developed in this work, can also be used to design more effective HWC systems in future field work.

## 6 Conclusion

We have employed a simple adaptation of 2D analytical method to model dynamic strain generated in HWC due to plane compressional waves propagation in multilayer media. This analytical model is useful to model simple scenarios and can validate boundary conditions applied in 2D and 3D numerical model of complex scenarios. To validate the choice of boundary conditions for 3D numerical model, strain values in helically-wound fiber optic cable estimated with our analytical method are compared to those modeled with numerical simulations. Despite its inherent approximation, the comparison of results obtained the analytical approach with 2D numerical modeling are quite acceptable and sufficient to confirm the accuracy of selected boundary conditions. We have investigated the effects of the surrounding media on the axial and radial strains of HWC for six scenarios representative of realistic situation and based on parameters of DAS experiment conducted at the New Afton deposit, British Columbia. Based on our parameters, our 3D numerical results show that the quality of cement (hard vs soft) between the casing and rock formation has a moderate effect on HWC data, with lower strain values observed for hard cement. However, having fully cured and emplaced cement inside and outside of casing is crucial to acquire decent signal from HWC. Effects of rock formation (hard vs soft) is more significant and can contribute to signal reduction on DAS data for HWC placed in soft rock formation. In all scenarios, effects are largest at higher incident angles (i.e. 90°). The presence of a water domain in the surrounding materials would make the fiber response more complicated owing to the combined effect of compressional and surface waves (i.e. tube waves). To this end, the presence of tube waves on HWC data at New Afton was beneficial to identify poorly cemented areas between the casing and rock formation, something that could not be unequivocally accomplished using straight fibre-optic cable alone.

**Conflict of interest**

The authors know of no conflicts of interest with this publication.

**Acknowledgments**

The authors would like to thank the Petroleum Technology Research Centre and Mitacs for funding in support of the modeling components of this research.

**Data Availability Statement**

The data that support the findings of this study are available from the corresponding author upon request.

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

**Appendix A. A 2-D analytical approach to determine the dynamic strain of helically wound cable**

For a two-dimensional (2-D) model, it is assumed that the seismic source is located far away from the location of the HWC.

This assumption results in a plane wave propagating in the in X-Z plane (Figure A.1). The governing equation for a plane sinusoidal wave in two-dimensional space is defined by following potential function ($\varphi$):

$$\varphi = A_0 exp\,(i(K_x * x + K_z * z - \omega t)) \tag{A.1}$$

Where $A_0$ is the initial amplitude of the seismic wave, $k_x$ is the wavenumber in the $x$ direction ($rad/m$), $K_z$ is the wavenumber in the $z$ direction ($rad/m$), $\omega$ is the angular frequency ($rad/s$), and $t$ is the time (s).

The following relationship exists between displacement (u) and the potential function:

$$u = \nabla_\varphi \tag{A.2}$$

$\nabla$ is is the collection of all the potential function's partial derivatives into a vector.

In order to define the compressional wave propagation equation as a function of incident angle (denoted by γ), we define the direction of ray path as a function of wave number (Fig. A.1).

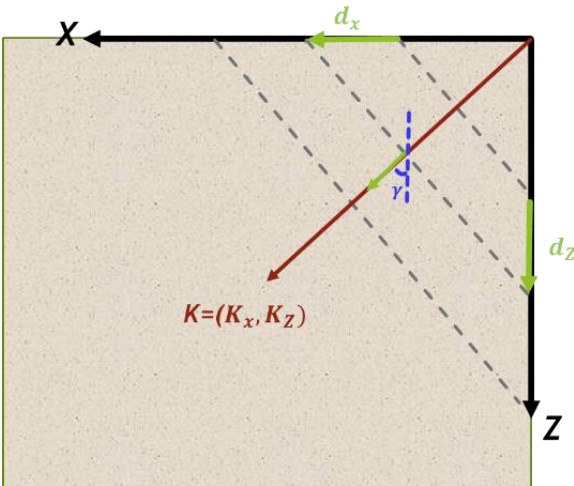

**Figure A.1: The arrow is used to denote a ray and the dashed line is used to denote a wavefront. $K$ indicates the direction of the ray. The angle $\gamma$ is the incident angle**

According to Fig. A.1, the wavenumber can be defined as follows:

$$k = k0(-\sin(\gamma), -\cos(\gamma)) \tag{A.3}$$

Where $k0$ is the initial wavenumber related to the initial source and $\gamma$ is the incident angle the wavefront relative to the z-axis. In the above equation, the negative signs represent propagation in the negative quadrant of the coordinate system.

Under ideal assumptions (e.g., uniform density, uniform tension, no resistance to motion, small deflection, etc.), one can show that the displacement ($u$) satisfies the two-dimensional semi-infinite wave equation with free ends:

$$u_{tt} = C^2 \nabla^2 u = C^2 (u_{xx} + u_{yy}) \tag{A.4}$$

Where $C$ is a fixed non-negative real coefficient, and $u_{tt}$ is $\frac{\partial^2 u}{\partial t^2}$

Assuming perfect plane wave propagation (it means that Because the source is far away, the spherical head wave is treated as a plane wave. As a result, displacement curve is in the plane), stresses generated inside the medium far away from the acoustic
source are obtained with the following equations:

$$\sigma_{xx} = (\lambda + 2\mu)\frac{\partial u_x}{\partial x} + \lambda \frac{\partial u_z}{\partial z} \tag{A.5}$$

$$\sigma_{zz} = (\lambda + 2\mu)\frac{\partial u_z}{\partial z} + \lambda \frac{\partial u_x}{\partial x} \tag{A.6}$$

$$\sigma_{xz} = \mu(\frac{\partial u_x}{\partial z} + \frac{\partial u_z}{\partial x}) \tag{A.7}$$

Where $\mu$ and $\lambda$ are shear modulus and lame constant, and $\sigma_{xx}$ and $\sigma_{zz}$ are stresses in x and z directions, respectively. By substituting equations A.1 and A.2 into A.5, A.6 and A.7:

$$\sigma_{xx} = A_0\big((\lambda + 2\mu)(k_x^2) + \lambda(k_z^2)\big)exp\big(i(k_x * x + k_z * z - \omega t)\big) \tag{A.8}$$

$$\sigma_{zz} = A_0\big((\lambda + 2\mu)(k_z^2) + \lambda(k_x^2)\big)exp\big(i(k_x * x + k_z * z - \omega t)\big) \tag{A.9}$$

$$\sigma_{xz} = A_0(2\mu k_x k_z)exp\big(i(k_x * x + k_z * z - \omega t)\big) \tag{A.10}$$

A plane acoustic wave is assumed to strike a multilayered media as shown in Figure A.2. The compressional wave originates
from a source located in layer 1 and proceeds through the intervening layers to emerge into layer n+1. It should be noted that the HWC is in layer $\frac{n}{2} + 1$. The layers are assumed to have infinite extent in the z direction, and the plane wave incident upon layer 1 is assumed to lie in the X-Z plane, making the problem two-dimensional. In effect, concentric layers that would exist

around a borehole in the true three-dimensional problem are represented as planar layers in this two-dimensional representation.

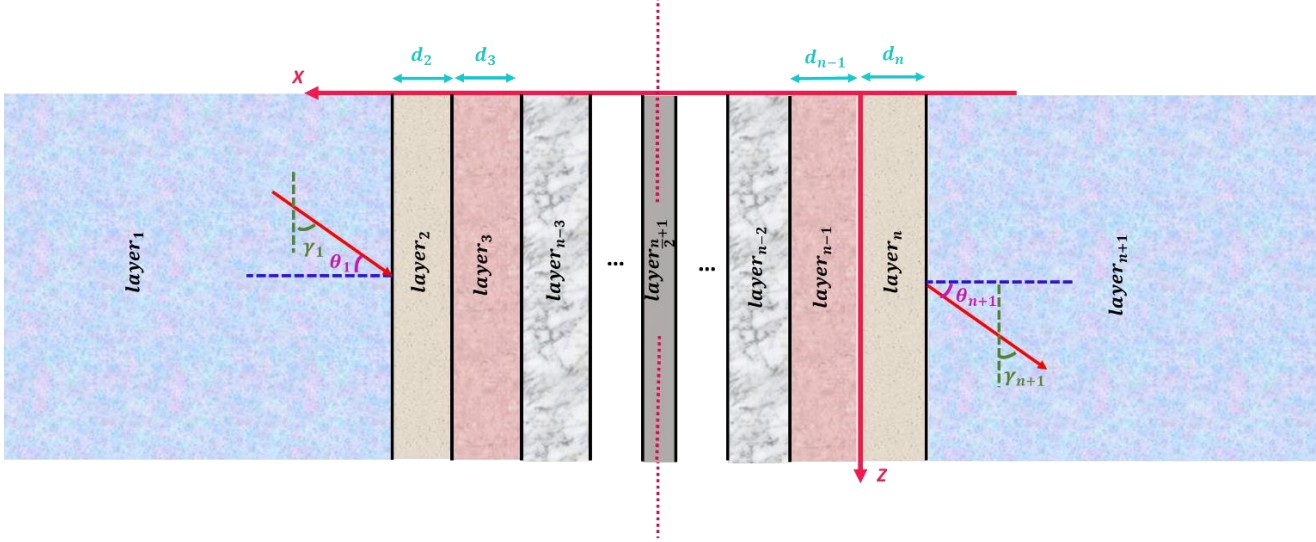


**Figure A.2. Propagation of compressional waves from source (layer 1) to layer n+1 (After Folds and Loggins, 1977). In the scenario considered in this work, layer n/2+1 represents the borehole, which is a plane of symmetry in this 2D plane-strain model (Each layer's thickness is represented by the letter d Layer number).**

When the stratified medium consists of parallel solid plates, the equations for one section must be related to those of the

adjacent section by proper boundary conditions used at interfaces between layers to satisfy continuity of normal ($\sigma_{xx}$) and shear stresses ($\sigma_{xz}$), and normal ($u_x$) and tangential displacements ($u_z$). Each layer (with layer number denoted by the index $l$) is considered linear elastic and isotropic if the following boundary conditions are satisfied:

$$u_x^l = u_x^{l+1} \tag{A.11}$$
$$u_z^l = u_z^{l+1} \tag{A.12}$$
$$\sigma_{xx}^l = \sigma_{xx}^{l+1} \tag{A.13}$$
$$\sigma_{xz}^l = \sigma_{xz}^{l+1} \tag{A.14}$$

At any layer, the displacements and stresses are given in terms of the following potential function:

$$\varphi_l = (A_l \, exp(i\alpha_l x) + B_l \, exp(-i\alpha_l x)) exp \, (i(\beta_l z - \omega t)) \tag{A.15}$$

Where $\varphi_l$ is the potential function for longitudinal waves, $\alpha_l$ is the x component of wavenumber, $\beta$ is the z component of

wavenumber, $A_l$ is the amplitude of incident wave, and $B_l$ the amplitude of reflected wave.

In equation A.15, $\alpha_l = k_l \, sin\gamma_l$ and $\beta_l = k_l \, cos\gamma_l$ are the x and z components of the wave vector, respectively. It should be noted that $k_l$ is the wavenumber in layer $l$.

As shown in Figure A.2, the angle of the wave vector with the normal of the interface is $\theta$. The following relationship exists between the angle of the wavefront relative to the $z$-axis and the interface normal vector:

$$\gamma_1 = 90 - \theta_1 \tag{A.16}$$

This yields:

$$\alpha_l = k_l \sin(\gamma_l) = k_l \sin(90 - \theta_l) = k_l \cos(\theta_l) \tag{A.17}$$

$$\beta_l = k_l \cos(\gamma_l) = k_l \cos(90 - \theta_l) = k_l \sin(\theta_l) \tag{A.18}$$

Clearly, based on Snell's equation, we must have:

$$\tag{A.19}$$
$$k_1 \sin\theta_1 = k_2 \sin\theta_2 = \cdots = k_{n-1} \sin\theta_{n-1} = k_n \sin\theta_n$$

As shown in equation A.15, $\beta$ is the same (according to Snell's law) for all layers. $\theta_l$ is the angle between the direction of the plane wave in layer $l$ and the normal of interface.

At any point inside or on the boundaries of layer $l$, the displacements and stresses are defined as follows:

$$u_x^l = \frac{\partial \varphi_l}{\partial x} \tag{A.20}$$

$$u_z^l = \frac{\partial \varphi_l}{\partial z} \tag{A.21}$$

$$\sigma_{xx}^l = \lambda \left( \frac{\partial u_x^l}{\partial x} + \frac{\partial u_z^l}{\partial z} \right) + 2\mu \frac{\partial u_x^l}{\partial x} \tag{A.22}$$

$$\sigma_{xz}^l = \mu \left( \frac{\partial u_x^l}{\partial z} + \frac{\partial u_z^l}{\partial x} \right) \tag{A.23}$$

The displacements and stresses on the right-side interface of layer n+1 are obtained by substituting equation A.15 into equations A.22 and A.23 and using $= n$ , $x = h_n + \epsilon$ (where $\epsilon$ is infinitesimal):

$$h_n = d_1 + d_2 + \cdots + d_{n-1} + d_n \tag{A.24}$$

This yields:

$$\begin{bmatrix} \sigma_{xx}^{n+1} \\ \sigma_{xz}^{n+1} \end{bmatrix} = \begin{bmatrix} G_{n+1} N_{n+1} & G_{n+1} M_{n+1} \\ R_{n+1} N_{n+1} & -R_{n+1} M_{n+1} \end{bmatrix} \begin{bmatrix} A_{n+1} \\ B_{n+1} \end{bmatrix} \tag{A.25}$$

with

$$N_{n+1} = \exp\left(i(\alpha_{n+1} h_n + \beta * z)\right) \tag{A.26}$$

$$M_{n+1} = \exp\left(i(\beta * z - \alpha_{n+1} h_n)\right) \tag{A.27}$$

$$R_{n+1} = -2\,\mu_{n+1}\,\alpha_{n+1}\beta \tag{A.28}$$

$$G_{n+1} = \left[(\lambda_{n+1} + 2\mu_{n+1})(-\alpha_{n+1}{}^2) + \lambda_{n+1}(-\beta^2)\right] \tag{A.29}$$

Now just inside the n$^{\text{th}}$ layer (e.g., at $= h_n - \epsilon$), the stresses are given as follows:

$$\begin{bmatrix} \sigma_{xx}^{n} \\ \sigma_{xz}^{n} \end{bmatrix} = \begin{bmatrix} G_n\,N_n & G_n\,M_n \\ R_n\,N_n & -R_n\,M_n \end{bmatrix} \begin{bmatrix} A_n \\ B_n \end{bmatrix} \tag{A.30}$$

with

$$N_n = \exp\left(i(\alpha_n h_n + \beta * z)\right) \tag{A.31}$$

$$M_n = \exp\left(i(\beta * z - \alpha_n h_n)\right) \tag{A.32}$$

$$R_n = -2\,\mu_n\,\alpha_n\beta \tag{A.33}$$

$$G_n = \left[(\lambda_n + 2\mu_n)(-\alpha_n{}^2) + \lambda_n(-\beta^2)\right] \tag{A.34}$$

Values in equation A.30 were obtained by substituting n=1 in equations A.17 and A.18.

Solving for the matrix of amplitude coefficients ($A_{n+1}$ and $B_{n+1}$):

$$\begin{bmatrix} A_{n+1} \\ B_{n+1} \end{bmatrix} = \begin{bmatrix} {}^{1}\!/{(2G_{n+1}N_{n+1})} & {}^{1}\!/{(2R_{n+1}N_{n+1})} \\ {}^{1}\!/{(2G_{n+1}M_{n+1})} & {}^{-1}\!/{(2R_{n+1}M_{n+1})} \end{bmatrix} \begin{bmatrix} \sigma_{xx}^{n+1} \\ \sigma_{xz}^{n+1} \end{bmatrix} \tag{A.35}$$

Based on the boundary conditions described in A.13 and A.14, and substituting equation A.30 into the right hand side of equation A.31, the following equation is obtained:

$$\begin{bmatrix} A_{n+1} \\ B_{n+1} \end{bmatrix} = \begin{bmatrix} {}^{1}\!/{(2G_{n+1}N_{n+1})} & {}^{1}\!/{(2R_{n+1}N_{n+1})} \\ {}^{1}\!/{(2G_{n+1}M_{n+1})} & {}^{-1}\!/{(2R_{n+1}M_{n+1})} \end{bmatrix} \begin{bmatrix} G_n\,N_n & G_n\,M_n \\ R_n\,N_n & -R_n\,M_n \end{bmatrix} \begin{bmatrix} A_n \\ B_n \end{bmatrix} \tag{A.36}$$

As a result, the following relationship exists between the amplitude of layer n+1 and layer n:

$$\begin{bmatrix} A_{n+1} \\ B_{n+1} \end{bmatrix} = \begin{bmatrix} \left({}^{N_n}\!/{2N_{n+1}}\right)\left({}^{G_n}\!/{G_{n+1}} + {}^{R_n}\!/{R_{n+1}}\right) & \left({}^{M_n}\!/{2N_{n+1}}\right)\left({}^{G_n}\!/{G_{n+1}} - {}^{R_n}\!/{R_{n+1}}\right) \\ \left({}^{N_n}\!/{2M_{n+1}}\right)\left({}^{G_n}\!/{G_{n+1}} - {}^{R_n}\!/{R_{n+1}}\right) & \left({}^{M_n}\!/{2M_{n+1}}\right)\left({}^{G_n}\!/{G_{n+1}} + {}^{R_n}\!/{R_{n+1}}\right) \end{bmatrix} \qquad \blacksquare$$

$$\times \begin{bmatrix} A_n \\ B_n \end{bmatrix} \tag{A.37}$$

The transformation matrix is named as $E_l$:

By considering:

$$\left(N_n/N_{n+1}\right) = exp\big(ih_n(\alpha_n - \alpha_{n+1})\big) = S_{n+1} \tag{A.38}$$

$$\left(N_n/M_{n+1}\right) = exp\big(ih_n(\alpha_n + \alpha_{n+1})\big) = V_{n+1} \tag{A.39}$$

$$\left(M_n/N_{n+1}\right) = exp\big(-ih_n(\alpha_n + \alpha_{n+1})\big) = W_{n+1} \tag{A.40}$$

$$\left(M_n/M_{n+1}\right) = exp\big(-ih_n(\alpha_n - \alpha_{n+1})\big) = J_{n+1} \tag{A.41}$$

$$E_{n+1} = \begin{bmatrix} (S_{n+1}/2)(G_n/G_{n+1} + R_n/R_{n+1}) & (W_{n+1}/2)(G_n/G_{n+1} - R_n/R_{n+1}) \\ (V_{n+1}/2)(G_n/G_{n+1} - R_n/R_{n+1}) & (J_{n+1}/2)(G_n/G_{n+1} + R_n/R_{n+1}) \end{bmatrix} = \begin{bmatrix} e_{11}^n & e_{12}^n \\ e_{21}^n & e_{22}^n \end{bmatrix} \tag{A.42}$$

$$\begin{bmatrix} A_{n+1} \\ B_{n+1} \end{bmatrix} = E_{n+1} \begin{bmatrix} A_n \\ B_n \end{bmatrix} \tag{A.43}$$

Application of the boundary condition at $x = h_n$ equates the velocities and stresses in layer n with those in layer n+1 at the interface, and the process used above can be repeated for each of the n-1 layers between n+1 and 1. Because of this, we can now write:

$$\begin{bmatrix} A_{n+1} \\ B_{n+1} \end{bmatrix} = [E_{n+1} E_n E_{n-1} \dots E_3 E_2] \begin{bmatrix} A_1 \\ B_1 \end{bmatrix} = \begin{bmatrix} F_{11} & F_{12} \\ F_{21} & F_{22} \end{bmatrix} \begin{bmatrix} A_1 \\ B_1 \end{bmatrix} \tag{A.44}$$

Where the F matrix represents the multiplication results ($F=[E_{n+1} E_n E_{n-1} \dots E_3 E_2]$).

We have assumed that the layer n+1 extends to infinity, which results in the fact that there are no reflections inside layer n+1 ($B_{n+1} = 0$). In equation A.44, $A_1$ is the amplitude of the wave generated by the seismic source and it is a known parameter. By having $B_{n+1} = 0$ and $A_1$, we can estimate $B_1$ and $A_{n+1}$ with the following equations:

$$B_1 = -(F_{21}/F_{22})A_1 \tag{A.45}$$

$$A_{n+1} = (F_{11})A_1 + (F_{12})B_1 \tag{A.46}$$

By knowing $A_1$ and $B_1$, it is possible to calculate the amplitude of incident and reflected waves inside each layer. As a result, by replacing these values inside equation A.15, we are able to calculate stresses and displacements for each layer.

Strains can be calculated directly from displacements with the following relationships:

$$e_{xx}^n = \partial u_{xx}^n/\partial x \tag{A.47}$$

$$e_{zz}^n = \partial u_{zz}^n/\partial z \tag{A.48}$$

Following the two-dimensional dynamic strain response on the HWC as represented by the coordinate system in the right-hand side of Fig. 2b, $e_{xx}$ is equal to radial strain ($e_{rr}$) and $e_{zz}$ is equal to longitudinal strain ($e_{zz}$) (Fig. 2b). As a result, fiber strain ($e_{||}$) is calculated as follows:

$$e_{||(Fiber)} = e_{zz(Cable)} \cos^2 \alpha + e_{rr(Cable)} \sin^2 \alpha \qquad (A.49)$$

## Appendix B. A pseudocode for 2-D analytical approach

 H=0;

**Foreach** $Theta_1 = 0 \dots M$ **do**

**Foreach** *layer n=1…n* **do**

$$K_n = \frac{(2 * pi * Freq)}{C_{p_n}}$$

$$Theta_n = \text{asin} \left(\left(\frac{K_n}{K_1}\right) * \sin(Theta_1)\right)$$

 $Alpha_n = K_n * \cos(Theta_n)$

$Beta_n = K_n * \sin(Theta_1)$

**If** $n == \left(\frac{n+1}{2}\right)$ **then**          // The Cable is located in the layer of ((n+1)/2)

$H_n = H + Diameter_n$ **else**

$H_n = H + \frac{Diameter_n}{2}$

 **endif**

$N_n = \exp(1i * (H_n * Alpha_n + Beta_n * z))$

$M_n = \exp(1i * (-H_n * Alpha_n + Beta_n * z))$

$R_n = -2 * \mu_n * Alpha_n * Beta_n$

$G_n = (-(\lambda_n + 2 * \mu_n) * (Alpha_n\text{^}2))(-(\lambda_n) * (Beta_n\text{^}2))$

 **If** $n > 1$ **then**

$S_n = \exp(1i * H_{n-1} * (Alpha_{n-1} - Alpha_n))$

$V_n = \exp(1i * H_{n-1} * (Alpha_{n-1} + Alpha_n))$

$W_n = \exp(-1i * H_{n-1} * (Alpha_{n-1} + Alpha_n))$

$J_n = \exp(-1i * H_{n-1} * (Alpha_{n-1} - Alpha_n))$

 $Array\ E_n[2][2]$

$$E_n[0][0] = \left(\frac{S_n}{2}\right) * \left(\left(\frac{G_{n-1}}{G_n}\right) + \left(\frac{R_{n-1}}{R_n}\right)\right)$$

$$E_n[0][1] = \left(\frac{W_n}{2}\right) * \left(\left(\frac{G_{n-1}}{G_n}\right) - \left(\frac{R_{n-1}}{R_n}\right)\right)$$

$$E_n[1][0] = \left(\frac{V_n}{2}\right) * \left(\left(\frac{G_{n-1}}{G_n}\right) - \left(\frac{R_{n-1}}{R_n}\right)\right)$$

$$E_n[1][0] = \left(\frac{J_n}{2}\right) * \left(\left(\frac{G_{n-1}}{G_n}\right) + \left(\frac{R_{n-1}}{R_n}\right)\right)$$

**Endif**

**End**

$$F = E_n * E_{n-1} * \ldots * E_2$$

$$B_1 = -\left(\frac{F[1][0]}{F[1][1]}\right) * A_1 \qquad \text{// } B_1 \text{ is the amplitude of reflected wave in the first layer}$$

$$A_n = F[0][0] * A_1 + F[0][1] * B_1 \qquad \text{// } A_n \text{ is the amplitude of transmitted wave in the last layer}$$

$$L = E_{(n+1)/2} * E_{(\frac{n+1}{2})-1} * \ldots * E_2$$

$$B_{(n+1)/2} = L[1][0] * A_1 + F[1][1] * B_1 \quad \text{// } B_{(n+1)/2} \text{ is the amplitude of reflection wave in the cable layer}$$

$$A_{(n+1)/2} = L[0][0] * A_1 + F[0][1] * B_1 \quad \text{// } A_{(n+1)/2} \text{ is the amplitude of refraction wave in the cable layer}$$

$$e_{xx(n+1)/2} = (Alpha_{(n+1/2)} \quad \wedge \quad 2) * \left(-A_{\frac{n+1}{2}} * \exp\left(1i * Alpha_{(n+1/2)} * H_{(n+1/2)}\right) - B_{\frac{n+1}{2}} * \exp\left(-1i * Alpha_{(n+1/2)} * \right.\right.$$

$$\left.\left. H_{(n+1/2)}\right)\right) * \exp\left(1i * Beta_{(n+1/2)} * z\right) \qquad \text{// } e_{xx(n+1)/2} \text{and } e_{zz(n+1)/2} \text{ are calculated strains in the cable layer}$$

$$e_{zz(n+1)/2} = -(Beta_{(n+1/2)} \quad \wedge \quad 2) * \left(A_{\frac{n+1}{2}} * \exp\left(1i * Alpha_{(n+1/2)} * H_{(n+1/2)}\right) + B_{\frac{n+1}{2}} * \exp\left(-1i * Alpha_{(n+1/2)} * \right.\right.$$

$$\left.\left. H_{(n+1/2)}\right)\right) * \exp\left(1i * Beta_{(n+1/2)} * z\right)$$

**End**