# Peer review of "Investigation of the Effects of Surrounding Media on the Distributed Acoustic Sensing of a Helically-Wound Fiber-Optic Cable with Application to the New Afton Deposit, British Columbia"

_Solid Earth, 2020_

## Referee Comment (RC1) · Anonymous Referee #1 · 4 Jan 2021

The authors consider an important problem, but they address it not in the best way.

The authors claim that they have developed a new method to model strain generated in fiber optic cables by seismic waves. However, the procedure described in Appendix A is about 70 years old. It was introduced in papers by Thomson (J. Appl. Phys., 21, 89–93, 1950) and Haskell (Bull. Seism. Soc. Am., 43, 17–34, 1953), and then modified by Fuchs (J. Phys. Earth, 16, 27–41, 1968) and Kennett(Bull. Seism. Soc. Am., 64, 1685–1696, 1974) to improve the numerical stability. This is a common procedure, which is frequently used in geophysical studies. It can be applied to both plane layers

and cylindrical layers.

In their study, the authors model a cylindrical cable as a plane layer. This is a rough approximation, which would be justified if there are no better approaches. However, this is not the case. In particular, the paper by Kuvshinov (Geophysical Prospecting, 64, 671-688, 2016) explains how to evaluate cable strain in a layered cylindrical geometry. I do not understand why the authors introduce an inferior method, which is not simper but less accurate.

As the authors said, it is important to analyze the effect of the surrounding medium on fiber optic cables. Such an analysis will be of interest if it is done in cylindrical geometry. The current model presented by the authors is not novel and it is a step back compared to the state-of-the-art.

---

## Referee Comment (RC2) · Anonymous Referee #2 · 8 Jan 2021

The paper discusses the impact of the surrounding media around borehole(s) on the DAS response collected with Helically-Wound Cables. It is an important topic relevant to the journal. Scientific relevance is good but the current manuscript requires some significant changes in my opinion for the following reasons.

General comments.

GC1: What is new in this work is not clear enough. As currently written, it looks like the derived 2D analytical solution is the novel aspect. a) If correct, then what are the

difference(s) and advantage(s) in comparison with the method proposed by Kuvshinov (AppendixD 2016) must be clarified. b) If not correct, i.e. if the 2D analytical solution only aims at validating the 3D FE modeling set-up, then the whole part about the analytical solution and how/if it fits with the FE modeling could remain in the appendix. c) In both cases, I suggest for the 2D analytical solution source code to be provided in order to help reproducing the results. Also not everybody can use COMSOL so this could be a valuable contribution.

GC2: FE modeling set-up and results, the main purpose of the paper, are not well enough discussed. a) No description of Fig 12 to 17 is given, so what do they exactly tell us? b) How exactly is the fiber strain extracted from this? c) Fig 15 and 16 exhibits strong strain variation inside/around the cable itself, so what exactly does the HW fiber sense? d) What does the white color mean in those plots (fig 16 and 17)? e) Why are the fiber strain values a magnitude smaller in 3D than in 2D (e.g. fig 18 versus fig 9) f) Given that 3D simulation is performed, a 3D view would be welcome!

GC3: The focus is on the HWC response exclusively. It should be clarified (ideally demonstrated and/or simulated) why the overall discussion does not apply to straight cables.

GC4: Given that the whole motivation is to explain discrepancies between Straight and HWC of Figure 22, one could discuss the impact of the fact that the gauge length of X meters results in a shorter distance with respect to the wavelength for the HWC compared to the straight cable.

GC5: Introduction part about HWC: Additional references should be made, e.g. Ning (2016, 2018 and 2019), Innanen (2017, 2019) or Eaid (2018) about directionality, multi-component sensing and other properties of HWCs.

GC6: Too many figures: a) Fig1 is a screenshot of Hornmann's youtube video and brings too little in the context. b) Fig 3 to 6 could fit in a single one. The reading would be facilitated. c) Fig 7 and 8 are too redundant. Should be merged into one and font

size to be increased. d) Fig 9 to 11 could go in Appendix if their goal is only the FE set-up validation. e) Fig 11: Marker size probably inappropriate – where are the red curve(s)? f) Fig 12 to 17: those FE snapshots may be key results but are not even discussed in the manuscript.

GC7: Results and Discussion. How the FE modeling outputs connect to the observations in Figure 22 is globally difficult to follow and therefore is not very convincing in my opinion. One may wonder if the field data should be introduced much earlier in the paper to explain the motivation behind the choice of simulation range.

GC8: Only incident P waves have been simulated. Why? What about S and maybe Rayleigh?

GC9: Are the responses frequency dependent? You focused on the 100Hz to be consistent with the field data

Specific comments: a) Line 14: not correct: off-axial strain is partly detected. Only pure broadside is not. b) Line 24: coupling -is- achieved. c) Line 26: Could be applied – not would. d) Line 51: Missing reference about axial (in)sensitivity (e.g. Mateeva 2014). e) Line 58: Patent request HWC Den Boer publ. year is 2013. f) Line 59-61: HWC geometry are nicely explained in paper-form by Kuvshinov 2016 and others (Eaid and Innanen). Probably better than a YouTube reference. g) Line 118: It's Kuvshinov 2016 . h) Line 120-132: A table could be clearer. i) Line 158: -In- to be removed. j) Line 172: One -the- to be removed. k) Fig 18 caption: what's the definition of "cement quality"? l) Fig 18 caption: bracket missing. m) Line 226 and 312: what's the meaning of this? Why does the presence of water involve surface waves? you probably mean S-waves?

---

## Author Comment (AC1) · 8 Jan 2021

Response to Referee #1

We would like to thank the reviewer for his/her insightful comments on the manuscript. Bellow is our response to the issues raised in the review:

Analytical methods still play an important role in geophysics and are often used to assess the response to specific geological situations for a variety of geophysical methods. However, it has long been recognized that analytical methods cannot handle

the complexity of many realistic geological situations. This is why numerical modeling methods including Finite-element (FE) have been developed and used widely for several decades. In our paper, we chose three-dimensional FE modeling as the primary method to model near-borehole effects on distributed acoustic sensing data acquired with helically-wound fibre-optic cable. The three-dimensional FE modeling approach used in our paper is state-of-the-art and provides all the accuracy required to model effects on such cable for complex and realistic geological situations. FE modeling has no geometry restrictions (i.e., planar or cylindrical) and further allows analyzing the strain around the cable (Figures 12-17), something not easy to achieve with analytical methods. As clearly stated in our paper, the analytical method introduced in Appendix is only used to validate the choice of boundary conditions of finite-element modeling and is by no means the primary method that we are advocating for in our work. The method is indeed not new and is a simple adaptation of Kennett's method. This is something that we will clarify in the revised manuscript. Despite its inherent approximation, the comparison of results obtained with 3D FE modeling and the simple analytical method shown in the paper are quite acceptable and sufficient to confirm the choice of the boundary conditions. The analytical method in the appendix is simpler to implement than the approach of Kuvshinov (2016) especially when a larger number of layers are surrounding the fibre-optic cable. This is why we chose it over other methods. Again, we would like to re-emphasize that the main results of our paper are those obtained with 3D FE modeling and that the analytical was used to confirm the choice of boundary conditions for the FE modeling.

---

## Author Comment (AC2) · 15 Jan 2021

We would like to appreciate the reviewer for her/his thoughtful comments and efforts towards improving our manuscript.Bellow is our response to the issues raised in the review:

GC 1: The paper aims at presenting FE modeling results showing the effect of various geological scenarios on DAS data acquired with a helically wound cable. The various scenarios were chosen to help assess possible causes of weak-amplitude data acquired at the New Afton mine using a HWC cable. We will clarify this in the revised manuscript. In the revised version, the code will be provided, and the structure of appendix will be modified according to the referee suggestion.

GC 2: a and b) In Fig 12 to 17, the radial strain around the cable as a response of compressional wave propagation in 90° incident angle is shown. In this FE modelling, the HWC fiber strain is calculated based on radial and axial strain of cable by considering wrapping angle ($\alpha$) equal to 30 degree, according to the following equation:

e_(HWC fiber)=e_axial*ãĂŰsinãĂŮˆ2 $\alpha$+e_radial*ãĂŰcosãĂŮˆ2 $\alpha$

c and d) Indeed, the cable is not exhibiting higher strain in these scenarios. As the number of domains is higher in other scenarios and smaller mesh are assigned to the thinner domains, COMSOL resolution is not able to show these strains and that is why cable domain is shown in white in other scenarios.

e) In 3D simulation, the real dimension of geometries is taken into account, and the cable is completely bounded by surrounding material and its radial deformation is controlled by its characteristics and surrounding material. While in 2-D simulation, one dimension (y) is not considered into modelling (plain strain assumption) and whatever energy is applied by plane wave is used to make deformation in x and z direction.

f) A 3D modelling figure will be presented in revised version.

GC3: This is correct; the focus is on the HWC response only. The response of straight fibre-optic cable is well-documented in the literature (Mateeva et al. 2014, Kuvshinov 1996). Straight fibre-optic cable are sensitive to the axial strain. For plane P-waves, the sensitivity of a straight fibre-optic cable varies as a function of cos2 $\theta$, $\theta$ being the angle between the plane wave direction and the cable. A HWC is sensitive to both axial and radial strain, the latter being dependent on the material around the cable. The FE modeling conducted in the paper aims primarily at recovering the radial strain averaged over the circumference of the cable. The response of the HWC is then obtained using

equation 1 above (see response to GC2). We will clarify this in the revised manuscript.

GC4: The gauge length is the physical interval along the optical fibre over which the difference between the phases of backscattered signal is measured and used to compute strain rate (Hartog 2018). The gauge length is the main factor controlling the spatial resolution. A trace spacing less than the gauge length is generally used during acquisition as data for overlapping gauge lengths generally improve the clarity of events, especially those related to slower waves (Hartog, 2018). The same gauge length (10 m) was used for both straight and helically-wound DAS data shown in figure 22. Due to the wrapping, strain rate over the gauge length is measured over a shorter cable distance for a helically-wound fibre than for a straight optic-fibre. This resulted in more channels for the HWC data (925 for HWC vs 813 for straight fibre-optic) for cables with identical length (828.5 m). While this is an important difference between straight and helically-wound cables, it does not explain the difference between data shown in figure 22 a and b. We will add this point to the discussion in the revised version.

GC 5: Mentioned additional references will be added in revised version.

GC 6: (a), (b), (c), and (d), we will consider those comments in the revised version.

e) They are tightly overlapped; it is the reason that the slight difference is not nicely detectable.

f) More discussion will be provided in revised version, regarding fig 12 to 17. Other sections of this comment will be applied in revised version.

GC 7: We will introduce the field data earlier in the paper to improve rationale for the choice of simulated models.

GC 8: We simulated incident P-waves because it is the primary body wave used for exploration purposes. S-waves are seldom used for exploration and are usually removed from the seismic data during data processing (at least direct and refracted S-waves). Rayleigh waves only propagate along a free surface and are not suitable source waves

for the exploration of the subsurface.

GC 9: Since the maximum response of fiber optic as a function of incident angle was the main objective, simulation was conducted in the frequency domain. We do not expect significant variations over seismic frequencies.

Specific comments: (a)-(j): We will consider those comments in the revised version.

k) The quality of cement is defined by Young's modulus, density, and Poisson's ration. In this study, what distinguishes between different cements are Young's modulus and density, and different velocities. The soft and hard cement have the same Young's modulus and Poisson's ratio as the cements used in Kuvshinov (2016).

(L) We will correct it in the revised version.

(m) Interface between water domain and solid domain acts as a free surface. When a head wave hits this interface, some part of energy converts to surface waves (tube waves) and the other part as compressional wave. Fluid does not support shear waves.

References: Hartog, A.H., 2018. An introduction to distributed optical fibre sensing. CRC Press. Kuvshinov, B.N., 2016. Interaction of helically wound fibre-optic cables with plane seismic waves. Geophysical Prospecting 64, 671–688. Mateeva, A., Lopez, J., Potters, H., Mestayer, J., Cox, B., Kiyashchenko, D., Wills, P., Grandi, S., Hornman, K., Kuvshinov, B., Berlang, W., Yang, Z., and Detomo, R., 2014. Distributed Acoustic Sensing for reservoir monitoring with vertical seismic profiling. Geophysical Prospecting 62, 679–692.

---

## Author Response (AR1)

Dear Reviewers,

We would like to thank you for your insightful comments on the manuscript. Below is our response to the issues raised in the review by indicating the lines in the track-file revised manuscript:

**Reviewer 1**

Response: Analytical methods still play an important role in geophysics and are often used to assess the response to specific geological situations for a variety of geophysical methods. However, it has long been recognized that analytical methods cannot handle the complexity of many realistic geological situations. This is why numerical modeling methods including Finite-element (FE) have been developed and used widely for several decades. In our paper, we chose three-dimensional FE modeling as the primary method to model near-borehole effects on distributed acoustic sensing data acquired with helically-wound fiber-optic cable. The three-dimensional FE modeling approach used in our paper is state-of-the-art and provides all the accuracy required to model effects on such cable for complex and realistic geological situations. FE modeling has no geometry restrictions (i.e., planar or cylindrical) and further allows analyzing the strain around the cable, something not easy to achieve with analytical methods. As clearly stated in our paper, the analytical method introduced in Appendix is only used to validate the choice of boundary conditions of finite-element modeling and is by no means the primary method that we are advocating for in our work. The method is indeed not new and is a simple adaptation of Kennett's method. This is something that we clarified in the revised manuscript. Despite its inherent approximation, the comparison of results obtained with 3D FE modeling and the simple analytical method shown in the paper are quite acceptable and sufficient to confirm the choice of the boundary conditions. The analytical method in the appendix is simpler to implement than the approach of Kuvshinov (2016) especially when a larger number of layers are surrounding the fiber-optic cable. This is why we chose it over other methods. Again, we would like to re-emphasize that the main results of our paper are those obtained with 3D FE modeling and that the analytical was used to confirm the choice of boundary conditions for the FE modeling.

Point-by-Point relevant change in the revised manuscript: Lines 30, 79, 82, 86-90, 116-121, 125-132, 405-411
* * *
**GC 1:** The paper aims at presenting FE modeling results showing the effect of various geological scenarios on DAS data acquired with a helically wound cable. The various scenarios were chosen to help assess possible causes of weak-amplitude data acquired at the New Afton mine using a HWC cable. We clarified this in the revised manuscript (Lines 30, 79, 82, 86-90, 116-121, 125-132, 405-411).

In the revised version, the source code is provided, and the structure of appendix was modified according to the referee suggestion (Lines 630-670, 115-121 )

**GC 2:**

a and b) The radial strain around the cable as a response of compressional wave propagation in 90° incident angle is shown (Lines 320-340). In this FE modelling, the HWC fiber strain is calculated based on radial and axial strain of cable by considering wrapping angle ($\alpha$) equal to 30 degree, according to the following equation: (Lines 199-201, 249-252)

$$e_{HWC\ fiber} = e_{axial} * sin^2\alpha + e_{radial} * cos^2\alpha \qquad (1)$$

c and d) Indeed, the cable is not exhibiting higher strain in these scenarios. As the number of domains is higher in other scenarios and smaller mesh are assigned to the thinner domains, COMSOL resolution is not able to show these strains and that is why cable domain is shown in white in other scenarios. (Lines 252-275).

e) In 3D simulation, the real dimension of geometries is taken into account, and the cable is completely bounded by surrounding material and its radial deformation is controlled by its characteristics and surrounding material. While in 2-D simulation, one dimension (y) is not considered into modelling (plain strain assumption) and whatever energy is applied by plane wave is used to make deformation in x and z direction (Lines 288-290).

**GC3:**

This is correct; the focus is on the HWC response only. The response of straight fibre-optic cable is well-documented in the literature *(*Mateeva *et al.* 2014, Kuvshinov 1996). Straight fibre-optic cable are sensitive to the axial strain. For plane P-waves, the sensitivity of a straight fibre-optic cable varies as a function of $cos^2 \theta$, $\theta$ being the angle between the plane wave direction and the cable. A HWC is sensitive to both axial and radial strain, the latter being dependent on the material around the cable. The FE modeling conducted in the paper aims primarily at recovering the radial strain averaged over the circumference of the cable. The response of the HWC is then obtained using equation 1 above (see response to GC2)(Lines 354-359).

**GC4:**

The gauge length is the physical interval along the optical fiber over which the difference between the phases of backscattered signal is measured and used to compute strain rate (Hartog 2018). The gauge length is the main factor controlling the spatial resolution. A trace spacing less than the gauge length is generally used during acquisition as data for overlapping gauge lengths generally improve the clarity of events, especially those related to slower waves (Hartog, 2018). The same gauge length (10 m) was used for both straight and helically-wound DAS data (Lines 364-367).

Due to the wrapping, strain rate over the gauge length is measured over a shorter cable distance for a helically-wound fibre than for a straight optic-fibre. This resulted in more channels for the HWC data (925 for HWC vs 813 for straight fibre-optic) for cables with identical length (828.5 m). While this is an important difference between straight and helically-wound cables, it does not explain the difference between data (Lines 434-440).

**GC 5:**

Mentioned additional references are added in revised version (Lines 458-459, 468-473, 486-490)

**GC 6:**

(a), (b), (c), and (d), we considered those comments in the revised version. (Lines 95, 190, 212, 245)

(e) They are tightly overlapped; it is the reason that the slight difference is not nicely detectable, however, we decided to delete that figure to make the story smoother and more integrated.

f) More discussion is provided in the revised version 17 (Lines 249-275)

**GC 7:**

We introduced the field data issue earlier in the paper to improve rationale for the choice of simulated models (Lines 86-93).

**GC 8:**

We simulated incident P-waves because it is the primary body wave used for exploration purposes. S-waves are seldom used for exploration and are usually removed from the seismic data during data processing (at least direct and refracted S-waves). Rayleigh waves only propagate along a free surface and are not suitable source waves for the exploration of the subsurface. (Lines 125-132).

**GC 9:**

Since the maximum response of fiber optic as a function of incident angle was the main objective, simulation was conducted in the frequency domain. We do not expect significant variations over seismic frequencies (Lines 133-138).

**Specific comments (mostly grammatical point):**

(a)-(d): We considered those comments in the revised version. (Lines 14, 29, 31, 57),

(e): Boer is for 2017, not 2013 that reviewer 2 mentioned)

(g): (Line 117)

(h): (Line 155-160)

(k) The quality of cement is defined by Young's modulus, density, and Poisson's ration. In this study, what distinguishes between different cements are Young's modulus and density, and different velocities. The soft and hard cement have the same Young's modulus and Poisson's ratio as the cements used in Kuvshinov (2016) (Lines 340-345).

(m) Interface between water domain and solid domain acts as a free surface. When a head wave hits this interface, some part of energy converts to surface waves (tube waves) and the other part as compressional wave. Fluid does not support shear waves (Lines 311, 420-422)

We were trying to cover all comments; we received from reviewers, however, in case of missing any point, I would appreciate if you would inform us about that.

If you have any additional questions/comments or further clarification, please do not hesitate to contact us.

Yours sincerely,

Mostafa

Mostafa Gorjian, BSc, MSc, PhD

Advanced Geological Engineering Lab.

Dept. of Earth, Ocean & Atmospheric Sciences | The University of British Columbia

2020-2207 Main Mall Vancouver, BC, Canada V6T 1Z4

Phone 604 827 1834, Cell 306 850 9349

mgorjian@eoas.ubc.ca

a place of mind
THE UNIVERSITY OF BRITISH COLUMBIA

---

## Editor Decision (ED1)

Comments on the manuscript of SE

Investigation of the Effects of surrounding media on the distributed acoustic sensing of a helically-wound fibre optic cable with application to the New Afton deposit, British Columbia.

By Hendi Sepidehalsadat et al.

**General comments**

Solid Earth is bunded to excellency of manuscripts before acceptance and the review process ensures the highest quality of the published papers. Checking the scientific quality of the final manuscript is the job of the editor. This manuscript received 2 reviews (referees 1 and 2) and a revised version was uploaded. The revised version was sent to referee 1 and an additional referee 3. The latest version of the manuscript has then been reviewed by myself (guest editor, see below details).

Following this latest review and following the referee's comments, the manuscript still cannot be accepted in its present form, as there are still many unclarities and inconsistencies in the manuscript, making your modelling comparison not reproducible. In addition, answers given in the real manuscript partially do not consider some referees comments, without clear enough justifications. There are also format problems in the presentations, e.g., in the abstract there should not be with paragraphs (as required by a referee), yet they are still present in the latest manuscript. One referee finds the abstract too long. In addition, minor elements such as double punctuation, unanswered response to the third reviewer, makes it as if, as suggested by the referee, corrections and preparation to the final version were too quickly implemented. Those formal issues could indeed be sorted out after the latest version has been presented and accepted, but they do not help convince us editors to accept the manuscript, especially a, and more importantly, the scientific content is still lacking consistency and information.

This is a pity as your work is very valuable for getting insights in the understanding of data recorded with helically wound fibre optic cable as compared to straight cables. It would indeed be great to provide to the community a simple method to understand helically wound fibre optic cable signals. Therefore, I suggest you address again and carefully previous remarks of the 3 referees, and implement them or not with clear justifications, and follow up with the concerns and the detailed issues indicated below, before the manuscript can be accepted. To do so, I would suggest you implement your answers within the referees' text and within this document, so it would be easier to follow up with the next steps, instead to have to jump from questions and answers, which does not help make the work easier. In addition, it is not sufficient to answer the referees and editor comments in such a file. The manuscript needs to change reflecting the comments, so that readers would not wonder with similar questions. This would help making your paper a cited paper.

If I understand correctly, the objective of your paper would be to make clear enough to readers that your simplified 2D solutions are close enough to 3D complex

modelling solutions and with which uncertainty. This simplified method, certainly easier to implement, would make indeed a valuable approach for understanding observations performed with a helicoidal cable (as mentioned by a referee). If this is not the objective of the paper, it indicates that it is not clear at all!

The first concern is that the results do not demonstrate clearly enough that you have made a valid 2D approach to replace or approximate a 3D computation. Let's consider an approach with 3 points below:
1. Using Comsol Multiphysics which is a 3D finite element computation tool, a 3D modelling can be performed really in 3D with both an helicoidal fibre and a straight fibre showing the "real" difference between them.
2. With the 3D modelling tool, you are in an excellent position to justify and demonstrate the difference between 3D computations and simplified results obtained with the 2D approach you propose. I am not a specialist of modelling. However, I guess implementing the comparison only by claiming boundary conditions are similar is not enough, is it? If yes, then you have to indicate clearly this in the text "as usually done in fem modelling, etc…" giving references. Not all readers are specialists of fem modelling method and its modelling tricks.
3. The referee 1 complains about your analytic method, although Kushinov (2016) proposed an approximation method which seems to be more powerful and accurate that the one you propose. If you disagree and want to show that the simplification you bring is still valid, then you need to show this much better. Why not implement Kushinov method (which seems simpler) and then compare it with your results and the 3D modelling with fem?
This approach would make your manuscript much more convincing.

The second concern is about the order of your presentation. The abstract does not even mention New Afton Deposit. As suggested by a reviewer, I would start by presenting the data with the issue you address by modelling. Something like "We do not understand why we do not measure the same thing as in the data, so this makes us willing to make 3D modelling. As those are complex, we propose a simplified method and show that this method is valid within ??%, and compares by ??% with Kushinov method." Instead, your discussion introduces data, which is clearly not the place to present them. The discussion should be more perspective on the method, what it would bring etc… and enlarge to other locations. I would therefore start from the data, indicating that you have issues to understand why the helically-wound fibre gives different results that straight fibres, and that you want to address this by modelling. This would make the story clearer.

The third concern is related to the presentation of your method, in particular in the appendix A. Notations are not consistent throughout the presentation, as the reference system is changing along the demonstration making results very confusing. In addition, equations disregard $\sigma_{xz}$ at some place, but then it is introduced for the boundary conditions in which $\sigma_{zz}$ does not exist, without explanation. The key equation A47 is wrong and should be corrected. Many symbols are not explained, making the reading hard to follow, unclear and may lead the reader to confusion and be doubtful on your work. In addition, hypothesis of the validity of the method are not exposed clearly enough. For example, what are the wave length of the seismic waves you use in the modelling? Any limitations? All

those issues should be indicated to make the paper as clearly and self-consistent as possible. This is at present not the case.

The forth concern is that some assertion are simply wrong, for example, on the inability of straight fibre to measure correctly certain seismic waves, and on imaging capability of Rayleigh waves for exploration. In both cases, this seems to be a lack of understanding of the methods. There are many references in the scientific literature that demonstrate the use of surface waves to image the subsurface.

Finally, the paper does not show clearly what the benefit of your study. How should we use the code to obtain which accuracy? Some reviewer asked about the size of the model. No answer on this appears in the manuscript.

**Additional details.**

Line 47. For completeness, you should add seismology. The references you use are quite old, and new applications have appeared, showing that the sentence line 52 and 56 is simply not true anymore (see for example

Jousset, P., Reinsch, T., Ryberg, T., Blanck, H., Clarke, A., Aghayev, R., Hersir, G. P., Henninges, J., Weber, M., Krawczyk, C. Dynamic strain determination using fibre-optic cables allows imaging of seismological and structural features. *Nature Communications* **9**, 2509. DOI: doi.org/10.1038/s41467-018-04860-y (2018).

Lindsey, N.J., Dawe, C. T. and Ajo-Franklin, J.B. Illuminating seafloor faults and ocean dynamics with dark fiber distributed acoustic sensing. *Science* **366**, 6469, 1103-1107, DOI: 10.1126/science.aay5881 (2019).

Sladen, A., Rivet, D., Ampuero, J-P., De Barros, L., Hello, Y., Calbris, G. & Lamare, P. Distributed sensing of earthquakes and ocean-solid Earth interactions on seafloor telecom cables. *Nature Communications* **18**, DOI: 10.1038/s41467-019-13793-z (2019).

Walter, F., Gräff, D., Kindner, F., Paitz, P., Köpfli, M., Chmiel, M. and Fichtner, A. Distributed acoustic sensing of microseismic sources and wave propagation in glaciated terrain. *Nature Communication* **11**, 2436. Doi:10.1038/s41467-020-15824-6 (2020).

At line 56, thanks to wave conversion, seismic strain can also be detected for hydraulic fracturing. In real Earth, not only P wave are generated, but also shear waves and Rayleigh waves, which all can be recorded with DAS. This paragraph is very restrictive and omit recent progresses. This does not withdraw the very important advantages that the helically wound fibre optic cable brings. But as it is said in the text now, one has the wrong impression that only helically cable would save DAS measurements. When looking at the figure 16 in the last part of the current manuscript, it is not obvious that this is really the case…

Line 70. I do not understand why you do not start from the observations from Figure 16, which is not new, so you could start from this.

In the figure 1, the scale is missing. How thick is the cable? $e_{zz}$ is certainly a very bad choice for expressing what by the way?? In Kushinov, this figure report $e_z$ for this term. Then in the appendix A, one is confused with the strain component. The issue, also in Kushinov to be fair, is that the reference you are using is not given. It would be nice to define all the axis x,y,z in a reference system that is consistent all along the demonstration. I would call this term $e_l$, as along the fibre, which then would allow you to call $e_{i\vee ii}$ as $e_{zz}$, along the vertical axis, much more conventional and which also can be used for the straight fibre.

Line 96. I start right away to be confused, as the reference given computes the transmission of waves in a medium made of parallel geological layers. It is not clear if you use the technique, but there is a flaw, as the cable "media" surrounding the fibre are not planes but cylinders, or is you use a geologically layered model in which the cable is drilling. How much error do you do when approximating a cylinder with a 2D model?

Line 104. It appears that your manuscript is based on the simple idea that "As boundary conditions give similar results with two methods (one simple 2D and one complex 3D), then I can use the 2D method in all cases". If you find a method which is simpler and more accurate, this is very good. However, it is not clear if the determination of the similarity in the boundary conditions in different methods is sufficient to validate the simplest approach in all case and inside the domain. If this is the case, then it must be made much clearer to the reader why the argumentation of saying "I can use a simpler 2D method instead of the more complex 3D method" is valid within this uncertainly level. However, at present due to the confusion introduced by the reading of the appendix a, then, your whole argumentation is very weak. May be elements of answers can be found in https://journals.sagepub.com/doi/full/10.1177/1094428116641191?

Line 110. Need to know more about this lucky P-wave which is working well with a simple model using a complex modelling tool... (velocity, source location, ...). How big is the model? I note here that a referee required that information, but they are missing in the current manuscript.

Line 111-115. Those are very strong limitations of your models, if they were true. I do not get the point of minimizing S waves and Rayleigh waves as if they would not be useful for exploration, just because you limit your study to P-waves. In addition, it is completely wrong to claim that "Rayleigh waves are nor suitable to explore the subsurface". There are a huge amount of publications using Rayleigh waves (e.g., multichannel analysis of surface waves) to image the structure of the subsurface, at all scales, including DAS as well.

Lines 199-120. Unclear. What are the "range of scenarios"? range of frequencies? What is typical for seismic frequencies?

Line 123. Missing space between in and Table.

Line 133. The sentence has no verb.

In table 1, check the text of all lines and correct where inaccurate (line 5 5). It is a pity not to consider a case scenario where the cable would be located behind casing.

Line 140. Bottom page note. I am not sure this is allowed in Solid Earth. Check in the format requirement and comply to them.

Fig. 2. In all figures, texts are too small. If it remains so, the reader will not read your paper, as unclear. The caption is very enigmatic. What are #1 and #2 etc.?

Line 150-151. The paragraph is "numerical modelling" with full 3D capability. You refer here to the analytical solution of Appendix A to compute separately radial and axial strain. I miss why you need to compute separately with two approximate methods, although you have a capability to compute in 3D. This actually, as suggested by one referee: did you actually made one full 3D computation? Why not show a 3D plot of the total strain around the 3d fibre? This is what we would like to see. This would be the reference for all 2D methods (yours and Kushinov, 2016) and plot the differences between such strains.

§2.3.  boundary conditions. It seems this is the core of the demonstration. This point should be made very clear: how a 3D cylindrical model is equivalent to a 2D flat model? Is it true that if I apply similar boundary conditions to the 2D as a simplified 3D or to a 3D model, the results will be the same?

In Figure 4, how were computed the numerical solutions? 2D or 3D? it is not clear what was the wrapping angle of the fibre you used in this model. It seems the solution will depend on it no?

Line 192-194. There is a logical issue. Comsol is able to compute 3D solutions directly for the full strain tensor. Why do you compute separately axial and radial? In addition, why do you need call to the 2D analytical solution? This is where we completely lose track of what you are really doing. In addition, before comparing several scenarios, we would like to see the differences between 2d and 3D for a similar configuration.

Line 196. It is not possible to judge for anyone if those are really rendering issues of the results. How to trust that the results are indeed valid in the cable? What makes you so confident? From the graphs (that I find too small once again), the mesh used seems very lose inside the cable. This is to me a critical issue, as we measure data in the cable. If modelling is not able to render what is happening in the cable, how can we envisage getting further in the use of your method for DAS? Getting those issue for the "rendering" rises also questions on the validity of the boundary conditions used, and your story collapses. In contrary, I would strongly suggest to give every effort to increase the resolution of the mesh to be able to compute properly the strain within the cable. I trust the mesh cell could be much smaller to image properly the fibre and all the layers surrounding it.

Line 203. Not clear if figure 6 is for hard? And fig 7 for soft? If yes, add "respectively".

Line 205. How do you see this result? We want to see much closer zoom.

Line 206. This is a real pity. And by the way, if this is computed correctly as you indicate at line 196, why can't you compare? Arguments on why it is not possible should be given, and I believe there is a way to compute with much higher resolution.

Line 218. I guess you mean figure 12 not 13.... In addition, which strain is shown? $\varepsilon_{xx}$ or $\varepsilon_{rr}$?

Line 225-230. Those lines reflect results that are not surprising. How could it be differently?

Line 235. Which frequency?

Line 236. Is attenuation considered? I understood that everything was elastic, actually acoustic, without S waves. So why mention attenuation? Or do you mean amplitude decrease due to geometrical spreading?

Line 242. Low incident angles... which values? where?

Line 248. "highly attenuating". This is not modelled, so irrelevant.

Line 250. Those waves are not surface waves, but interface waves.

Line 251. I do not understand this argument. It seems wrong. In #6, the maximum amplitude is higher than in #5 indeed, but lower than in other scenarios without water. Can you explain (in the manuscript as well)?

Figures 6-11 should display similar color scale and range so we can compare results.

Fig. 12.15 could be shown all together for easier comparison. I do not get the benefice of comparing 2 by 2.

Figure 7 to 11. "Radial". I really do not follow how the radial strain around the cable has this shape. Where is the wave coming from? Has the wave a waveform? The strain is changing with time, at which time is it represented? The scale between the different figure should be the same. The mesh seems better defined in Fig 8 and 9 compared to the figure 7-8 in the most central part. Why is this? Would it be possible to zoom closer to the cable to see the shape of the strain inside the cable? Figure 12. and 13. Indicate on the figure where is soft and hard. Reporting scenario # is nice, but not very clear enough. The caption could get the list of scenarios main characteristics this comment is not mandatory, but it would make the reading much simplified). I see on figure much less than 10% difference between the curves, but at line 221, there is another value indicated, make it consistent.

Figure 14. line 285. Rephrase: "strains" are not "angles".

I still do not understand why it is not possible to draw a figure with few loops of the 3d helix and show us how strain components vary along the fibre, from the 3D computation.

Discussion.
As presented here this is not a discussion. A discussion would make the possible limitation of your study, and how to improve them. It would also be the location to discuss how to improve the deployment. Do you have suggestions, etc…

Line 292. For completeness, I would also indicate Reinsch et al, 2017.
Reinsch T., Thurley T., Jousset P., 2017. On the mechanical coupling of a fiber optic cable used for distributed acoustic/vibration sensing applications — a theoretical consideration. - Measurement Science and Technology, 28, 12. http://doi.org/10.1088/1361-6501/aa8ba4

Line 295. This is not shown at all. You never made the link between your theoretical study and the new Afton experiment that we hear about now only. In addition, this is not the place in the discussion. Why not starting the paper from this observation and justify all efforts. The introduction would be much easier to set up, and this would give an objective clear for your study.

Line 301. Space missing between 10 m and for.

Line 310. Was the straight cable a Constellation fibre? And what about the helically one?

Line 370 – figure 16. What is shown? Longitudinal strain? Radial strain? What is the color map used?

The reference list is to be checked. For example, Innanen et al. 2019 is not cited in the manuscript.

Appendix A.
Line 441. You cannot start the appendix by "this". Which one?

Line 443. The potential function letter (phi) is missing.

Line 447. What is $\nabla$?

Line 453. You mean A.1?

Line 460. What is a "perfect" plane wave propagation?

Equations A5 and A6. Where is the shear stress component? Do you develop the method only for P-wave? This is very limiting… How would it be to expand to S-wave as well?

Line 467-469. This approximation is really strong. A better quantification on the error made should be properly given.

Line 470. You did not indicate what are the layers. Otherwise it is a bit more criptic...

Line 475-477. The shear stresses do exist! But where is $\sigma_{zz}$?

Line 477. What is this reference to "After 13"?? It seems you copied your text from somewhere without checking properly the meaning of what you write...

Line 488. Again $\sigma_{zz}$ is absent.

Line 490, why do you need the infinitesimal factor ?

Line 491. Define d1, d2, etc... from fig A.2?

Equation A42. not clear what F11, F12 etc... are.

Line 506. A1 is the amplitude of the wave generated by the seismic source. It is known in the case of an active experiment. However, one may want to use helically fibre with microseismic data, where the source is unknown... Are we then lost?

The term $\varepsilon_{xz} = \frac{1}{2}\left(\frac{\partial u_{xx}^{n}}{\partial z} + \frac{\partial u_{zz}^{n}}{\partial x}\right)$ is missing, why?

Line 513. This approximation is really the strong assumption of the approximation which allows you to migrate from 3D to 2D. This is exactly this difference we want to see between the two, and validated by the 3D true numerical computation.

As indicated by several referees, this equation is wrong sand should be corrected.

As you see, there are still many unclarities, questions, and I guess my reading was not exhaustive. I would suggest you to read the manuscript again very carefully and bring all attention to make the manuscript readable, clearer and fully justified following the indications given by the referee reports and this review.

Once those details will have been addressed, we can reconsider your manuscript.

---

## Author Response (AR2)

Dear Editor,

First and foremost, thank you for your helpful feedback and that of the reviewers. We made a concerted attempt to respond to the majority of the comments clearly and concisely. Specifically, we clarified that the simple 2D analytical method proposed in the Appendix is used to validate the boundary conditions for the 2D Finite-element method. Results from both 2D analytical and FE share the same geometry and thus, can be compared. Although more sophisticated, Kuvshinov's analytical approach was not used because it does not reproduce exactly the geometry and assumption of the 2D FE modelling (due to the averaging of the radial strain over the outer diameter of the cable). Boundary conditions determined in 2D were then adapted to 3D FE following symmetry and geometry considerations. In addition, we clarified why only P-waves were considered in both analytical and FE modeling. This has to do with the DAS-VSP data acquired at New Afton which only focused on P-wave reflections. As suggested, we moved the section introducing the New Afton data earlier in the manuscript. This helped clarify the context of our work and certain simplification/assumption made in our approach (i.e. P-waves only). We also clarified that DAS is by no means limited to P-wave exploration.

In the attached files, we provide a point-by-point response to comments raised by the reviewers and the editor. We hope that you will find changes applied to the manuscript and responses provided acceptable.

Thank you for considering our manuscript for publication in Solid Earth.

Sincerely,

On behalf of all the authors

Comments on the manuscript of SE

Investigation of the Effects of surrounding media on the distributed acoustic sensing of a helically-wound fibre optic cable with application to the New Afton deposit, British Columbia. By Hendi Sepidehalsadat et al.

**General comments**

Solid Earth is bunded to excellency of manuscripts before acceptance and the review process ensures the highest quality of the published papers. Checking the scientific quality of the final manuscript is the job of the editor. This manuscript

received 2 reviews (referees 1 and 2) and a revised version was uploaded. The revised version was sent to referee 1 and an additional referee 3. The latest version of the manuscript has then been reviewed by myself (guest editor, see below details).

Following this latest review and following the referee's comments, the manuscript still cannot be accepted in its present form, as there are still many unclarities and inconsistencies in the manuscript, making your modelling comparison not reproducible. In addition, answers given in the real manuscript partially do not consider some referees comments, without clear enough justifications. There are also format problems in the presentations, e.g., in the abstract there should not be with paragraphs (as required by a referee), yet they are still present in the latest manuscript. One referee finds the abstract too long. In addition, minor elements such as double punctuation, unanswered response to the third reviewer, makes it as if, as suggested by the referee, corrections and preparation to the final version were too quickly implemented. Those formal issues could indeed be sorted out after the latest version has been presented and accepted, but they do not help convince us editors to accept the manuscript, especially a, and more importantly, the scientific content is still lacking consistency and information. This is a pity as your work is very valuable for getting insights in the understanding of data recorded with helically wound fibre optic cable as compared to straight cables. It would indeed be great to provide to the community a simple method to understand helically wound fibre optic cable signals. Therefore, I suggest you address again and carefully previous remarks of the 3 referees, and implement them or not with clear justifications, and follow up with the concerns and the detailed issues indicated below, before the manuscript can be accepted. To do so, I would suggest you implement your answers within the referees' text and within this document, so it would be easier to follow up with the next steps, instead to have to jump from questions and answers, which does not help make the work easier. In addition, it is not sufficient to answer the referees and editor comments in such a file. The manuscript needs to change reflecting the comments, so that readers would not wonder with similar questions. This would help making your paper a cited paper.

If I understand correctly, the objective of your paper would be to make clear enough to readers that your simplified 2D solutions are close enough to 3D complex modelling solutions and with which uncertainty. This simplified method, certainly easier to implement, would make indeed a valuable approach for understanding

observations performed with a helicoidal cable (as mentioned by a referee). If this is not the objective of the paper, it indicates that it is not clear at all!

(It is explicitly stated in different locations what are the objectives of this paper, for example "we first present a simple adaptation of the 2-D analytical solution of Folds and Loggins (1977) to model HWC's dynamic strain due to acoustic waves. The analytical solution estimates the HWC response using a relatively simple model that does not require specialized software. In the context of this work, the most significant value of this

analytical solution is that it provides a means to validate the choice of boundary conditions of a 2-D numerical model developed using the commercial software COMSOL Multiphysics. Then, having established the effectiveness of the 2-D numerical model, a 3-D model was developed using the same methods and tools, by adapting 2D boundary conditions to 3D simulation based on geometrical and symmetry considerations. Material properties and HWC geometry used for the modeling were based on the field conditions and data previously acquired during a survey at the New Afton deposit in Canada. Several scenarios are modelled and compared to help identify near borehole conditions explaining the difference between data acquired with straight and HWC at New Afton, specifically the weak-amplitude data obtained with HWC (Figure 2). The HWC modeling results and approach used here can provide insights for optimizing field deployments. Service companies could benefit by using this workflow prior to installation of HWC fiber optic cable in the field")

The first concern is that the results do not demonstrate clearly enough that you have made a valid 2D approach to replace or approximate a 3D computation. Let's consider an approach with 3 points below:

1. Using Comsol Multiphysics which is a 3D finite element computation tool, a 3D modelling can be performed really in 3D with both an helicoidal fibre and a straight fibre showing the "real" difference between them.

2. With the 3D modelling tool, you are in an excellent position to justify and demonstrate the difference between 3D computations and simplified results obtained with the 2D approach you propose. I am not a specialist of modelling. However, I guess implementing the comparison only by claiming boundary conditions are similar is not enough, is it? If yes, then you have to indicate clearly this in the text "as usually done in fem modelling, etc…" giving references. Not all readers are specialists of fem modelling method

and its modelling tricks (Please see our response to the prior remark, which is marked in yellow.)

3. The referee 1 complains about your analytic method"), although Kushinov (2016) proposed an approximation method which seems to be more powerful and accurate that the one you propose. If you disagree and want to show that the simplification you bring is still valid, then you need to show this much better. Why not implement Kushinov method (which seems simpler) and then compare it with your results and the 3D modelling with fem?

This approach would make your manuscript much more convincing.

The 2D analytical method presented in the Appendix is used to validate the boundary conditions for 2D FE modeling. The 2D geometry of both analytical and FE simulation are identical, meaning that results can be compared. This is not the case with Kuvshinov's approach which includes averaging of the radial strain over the outter diameter of the cable, casing, etc. Thus, this is why Kuvshinov's method, although more sophisticated is not used. Once properly established in 2D, boundary conditions were then adapted to the 3D FE simulation based on geometry and symmetry relationships. The three-dimensional FE modeling approach used in our paper is state-of-the-art and provides all the accuracy required to model HWC embedded in complex and realistic geological situations. FE modeling has no geometry restrictions (i.e., planar or cylindrical) and further allows analyzing the strain around the cable, something not easy to achieve with analytical methods. Again, we would like to re-emphasize that the main results of our paper are those obtained with 3D FE modeling and that the analytical method is used to confirm the choice of boundary conditions for the FE modeling.

The second concern is about the order of your presentation. The abstract does not even mention New Afton Deposit (It did and still does mention New Afton). As suggested by a reviewer, I would start by presenting the data with the issue you address by modelling (Applied). Something like "We do not understand why we do not measure the same thing as in the data, so this makes us willing to make 3D modelling. As those are complex, we propose a simplified method and show that this method is valid within ??%, and compares by ??% with Kushinov method." Instead, your discussion introduces data, which is clearly not the place to present them (That part was moved to the introduction). The discussion should be more perspective on the method, what it would bring etc… and enlarge to other locations. I would therefore start from the data, indicating that you have issues to understand why the helically-wound fibre gives different results that straight fibres, and that you want to address this by modelling. This would make the story clearer. The third concern is related to the presentation of your method, in particular in the appendix A. Notations are not consistent (Fixed) throughout the presentation, as the reference system is changing along the demonstration making results very confusing. In addition, equations disregard σxz at some place, but then it is introduced for the boundary conditions in which σzz does not exist, without explanation (Applied). The key equation A47 is wrong (We think notation was confusing, the equation is still valid) and should be corrected. Many symbols are not explained, making the reading hard to follow, unclear and may lead the reader to confusion and be doubtful on your work. In addition, hypothesis of the validity of the method are not exposed clearly enough. For example, what are the wave length of the seismic waves you use in the modelling (Added)? Any limitations? All 2 those issues should be indicated to make the paper as clearly and self-consistent as possible. This is at present not the case. The forth concern is that some assertion are simply wrong, for example, on the inability of straight fibre to measure correctly certain seismic waves, and on imaging capability of Rayleigh waves for exploration. In both cases, this seems to be a lack of understanding of the methods. There are many references in the scientific literature that demonstrate the use of surface waves to image the subsurface (Fixed)

Finally, the paper does not show clearly what the benefit of your study. How should we use the code to obtain which accuracy? Some reviewer asked about the size of the model (Applied). No answer on this appears in the manuscript.

(The RMSE is in the range of 0.003-0.004ε, which is a negligible error when comparing the 2d numerical modelling with the analytical model. The results of this paper can be used as a guideline for analyzing the impact of surrounding media and incident angle on the response of helically wound cable, optimizing the installation of helically wound cable in various conditions, and to validate boundary conditions of 3-D numerical model built for analyzing complex scenarios)

**Additional details.**

Line 47. For completeness, you should add seismology.

 The references you use are quite old, and new applications have appeared, showing that the sentence line 52 and 56 is simply not true anymore (see for example Jousset, P., Reinsch, T., Ryberg, T., Blanck, H., Clarke, A., Aghayev, R., Hersir, G. P., Henninges, J., Weber, M., Krawczyk, C. Dynamic strain determination using fibre-optic cables allows imaging of seismological and structural features. *Nature Communications* **9**, 2509. DOI:

doi.org/10.1038/s41467-018-04860-y (2018). (Added)

Lindsey, N.J., Dawe, C. T. and Ajo-Franklin, J.B. Illuminating seafloor faults and ocean  dynamics with dark fiber distributed acoustic sensing. *Science* **366**, 6469, 1103-1107, DOI: 10.1126/science.aay5881 (2019). (Added)

Sladen, A., Rivet, D., Ampuero, J-P., De Barros, L., Hello, Y., Calbris, G. &

Lamare, P. Distributed sensing of earthquakes and ocean-solid Earth interactions on seafloor telecom cables. *Nature Communications* **18**, DOI: 10.1038/s41467- 019-13793-z (2019). (Added)

Walter, F., Gräff, D., Kindner, F., Paitz, P., Köpfli, M., Chmiel, M. and Fichtner, A. Distributed acoustic sensing of microseismic sources and wave propagation in glaciated terrain. *Nature Communication* **11**, 2436. Doi:10.1038/s41467-020-15824-6 (2020). (Added)

At line 56, thanks to wave conversion, seismic strain can also be detected for hydraulic fracturing. In real Earth, not only P wave are generated, but also shear waves and Rayleigh waves, which all can be recorded with DAS. This paragraph is very restrictive and omit recent progresses (Fixed). This does not withdraw the very important advantages that the helically wound fibre optic cable brings. But as it is said in the text now, one has the wrong impression that only helically cable would save DAS measurements. When looking at the figure 16 in the last part of the current manuscript, it is not obvious that this is really the case 3 (We clarified why only P-waves were considered in the modeling work. This is simply because the reflected wavefield at New Afton is dominated by P-waves. S-wave reflections and mode-converted waves (P-S and S-P) were not identified on that data. Surface waves were not observed on the data measured deep at the New Afton mine).

Line 70. I do not understand why you do not start from the observations from Figure 16, which is not new, so you could start from this (Fixed).

In the figure 1, the scale is missing. How thick is the cable? (this info is in Table 2)

*ezz* is certainly a very bad choice for expressing what by the way?? In Kushinov, this figure report *ez* for this term. Then in the appendix A, one is confused with the strain component. The issue, also in Kushinov to be fair, is that the reference you are using is not given. It would be nice to define all the axis x,y,z in a reference system that is consistent all along the demonstration. I would call this term *el*, as along the fibre, which then would allow you to call *e*.V. . as *ezz*, along the vertical axis, much more conventional and which also can be used for the straight fibre. (Applied)

Line 96. I start right away to be confused, as the reference given computes the transmission of waves in a medium made of parallel geological layers. It is not clear if you use the technique, but there is a flaw, as the cable "media" surrounding the fibre are not planes but cylinders, or is you use a geologically layered model in which the cable is drilling. How much error do you do when approximating a cylinder with a 2D model?

(Please refer to page 2, the text highlighted in gray)

Line 104. It appears that your manuscript is based on the simple idea that "As boundary conditions give similar results with two methods (one simple 2D and one complex 3D), then I can use the 2D method in all cases". If you find a method which is simpler and more accurate, this is very good. However, it is not clear if the determination of the similarity in the boundary conditions in different methods is sufficient to validate the simplest approach in all case and inside the domain. If this is the case, then it must be made much clearer to the reader why the argumentation of saying "I can use a simpler 2D method instead of the more

complex 3D method" is valid within this uncertainly level. However, at present due to the confusion introduced by the reading of the appendix a, then, your whole argumentation is very weak. May be elements of answers can be found in https://journals.sagepub.com/doi/full/10.1177/1094428116641191?

(Please check the yellow highlighted answer above and also "The modeling work presented in this paper is used to help understand the performance of a helically-wound fiber from a VSP field study at the New Afton mine (Bellefleur et al., 2020). Straight and helically-wound fibre optic cables were deployed in a single borehole to assess the efficiency of DAS to detect geological interfaces and structures associated with mineralization. The New Afton deposit is a porphyry deposit comprising primarily disseminated Cu-Au mineralization. The fibre-optic cables were deployed in a steeply-dipping (70° from horizontal) deviated borehole starting in a work bay located 650 m below the surface and ending at a depth of approximately 1300 m. Both cables were placed inside steel drill rods (used as casing) and cemented in place with grout. The grout was circulated to the bottom of the borehole via a grout tube located inside the casing until grout eventually reached surface from both inside and outside of the casing (drill rods). The grout cured for one month prior to the VSP survey. Based on the afore-noted grout returns both within and outside the casing, it was assumed at the time that grout had filled both the casing and the casing-formation annulus; however, data collected during the survey (discussed below) suggests this may not have been the case. The data were acquired with 1 kg of explosives fired in a 20 m deep shot hole at surface".)

Line 110. Need to know more about this lucky P-wave which is working well with a simple model using a complex modelling tool… (velocity, source location, …). How big is the model? I note here that a referee required that information, but they are missing in the current manuscript. (Applied)

Line 111-115. Those are very strong limitations of your models, if they were true. I do not get the point of minimizing S waves and Rayleigh waves as if they would not be useful for exploration, just because you limit your study to P-waves. In addition, it is completely wrong to claim that "Rayleigh waves are nor suitable to explore the subsurface". There are a huge amount of publications using Rayleigh waves (e.g., multichannel analysis of surface waves) to image the structure of the subsurface, at all scales, including DAS as well.

(This has been re-written to clarify the context of our work)

Lines 199-120. Unclear. What are the "range of scenarios"? range of frequencies? What is typical for seismic frequencies? (It was in frequency domain, with dominant frequency of 100Hz)

Line 123. Missing space between in and Table. (Applied)

Line 133. The sentence has no verb. (Fixed)

In table 1, check the text of all lines and correct where inaccurate (line 5 5). It is a pity not to consider a case scenario where the cable would be located behind casing. (Scenarios were consistent with the field tests Because the cable was positioned inside the casing in the New Afton mine, a response of fibre while cable is located behind the casing is not considered.)

Line 140. Bottom page note. I am not sure this is allowed in Solid Earth. Check in the format requirement and comply to them. (We saw bottom page notes in papers published in Solid Earth)

Line 150-151. The paragraph is "numerical modelling" with full 3D capability. You refer here to the analytical solution of Appendix A to compute separately radial and axial strain. I miss why you need to compute separately with two approximate methods, although you have a capability to compute in 3D. This actually, as

suggested by one referee: did you actually made one full 3D computation? Why not show a 3D plot of the total strain around the 3d fibre? This is what we would like to see. This would be the reference for all 2D methods (yours and Kushinov, 2016) and plot the differences between such strains.

(The purpose of using a 2D analytical method is to validate the choice of boundary conditions in numerical simulations. In this simulation, it is aimed to investigate the strain of cable in radial and axial directions and relate these strains to the strain of fiber, using equation #1 by assuming that the fibre is wrapped around the cable at an angle of 30 degrees. This wrapping angle is derived from the HWC cable used in the New Afton mine. As mentioned previously, the impact of the surrounding media is best observed on the radial strain. Thus, modelling strains separately allows to show details of radial strain and assess the effect of surrounding media more effectively).

§2.3. boundary conditions. It seems this is the core of the demonstration. This point should be made very clear: how a 3D cylindrical model is equivalent to a 2D flat model? Is it true that if I apply similar boundary conditions to the 2D as a simplified 3D or to a 3D model, the results will be the same?

(Because real geometry cannot be modelled in a 2d model, the results will vary based on the geometry, and if it is modelled in 2d, it will result in significant errors.)

In Figure 4, how were computed the numerical solutions? 2D or 3D? it is not clear what was the wrapping angle of the fibre you used in this model. It seems the solution will depend on it no?

(clarified in the text. The fibre is wrapped around the cable at an angle of 30 degrees. This wrapping angle is derived from the HWC cable used in the New Afton mine. Indeed, the response depends on the wrapping angle.)

Line 192-194. There is a logical issue. Comsol is able to compute 3D solutions directly for the full strain tensor. Why do you compute separately axial and radial? In addition, why do you need call to the 2D analytical solution? This is where we completely lose track of what you are really doing. In addition, before comparing several scenarios, we would like to see the differences between 2d and 3D for a similar configuration.

(The purpose of using a 2D analytical method is to validate the numerical simulations. In this simulation, it is aimed to investigate the strain of cable in radial and axial directions and relate these strains to the strain of fiber, using equation A.49 by assuming that the fibre is wrapped around the cable at an angle of 30 degrees. This wrapping angle is derived from the HWC cable used in the New Afton mine)

Line 196. It is not possible to judge for anyone if those are really rendering issues of the results. How to trust that the results are indeed valid in the cable? What makes you so confident? From the graphs (that I find too small once again), the mesh used seems very lose inside the cable. This is to me a critical issue, as we measure data in the cable. If modelling is not able to render what is happening in the cable, how can we envisage getting further in the use of your method for DAS? Getting those issue for the "rendering" rises also questions on the validity of the boundary conditions used, and your story collapses. In contrary, I would strongly suggest to give every effort to increase the resolution of the mesh to be able to compute properly the strain within the cable. I trust the mesh cell could be much smaller to image properly the fibre and all the layers surrounding it.

The picture below should clarify that there are no meshing issues with COMSOL. Inside the domain, a tetrahedral mesh was employed, while lower reflecting surfaces were covered with a swept mesh (see figure 4).

[Figure]

*Plan View*

*Side View*

• *This Type of Mesh is mapped mesh. It is strongly recommended to use mapped meshing in the infinite boundaries to prevent poor mesh element quality, giving rise to poor or slow convergence for iterative solvers and making the problem ill-conditioned in general.*

*Front View*

• *This Type of Mesh is mapped mesh. It is strongly recommended to use mapped meshing in the infinite boundaries to prevent poor mesh element quality, giving rise to poor or slow convergence for iterative solvers and making the problem ill-conditioned in general.*

Line 203. Not clear if figure 6 is for hard? And fig 7 for soft? If yes, add "respectively". (applied)

Line 206. This is a real pity. And by the way, if this is computed correctly as you indicate at line 196, why can't you compare? Arguments on why it is not possible should be given, and I believe there is a way to compute with much higher resolution.

(Applied)

Line 218. I guess you mean figure 12 not 13…. In addition, which strain is shown? $\varepsilon\, xx$ or $\varepsilon rr$?

(Applied)

Line 225-230. Those lines reflect results that are not surprising. How could it be differently?

(Because real geometry cannot be modelled in a 2d model, the results would vary based on the geometry)

Line 235. Which frequency? (applied)

Line 236. Is attenuation considered? I understood that everything was elastic, actually acoustic, without S waves. So why mention attenuation? Or do you mean amplitude decrease due to geometrical spreading?

Clarifed. Attenuation is not considered in our modeling. In addition, there is no geometrical spreading for plane waves. This is the definition for a point source: As the wavefront moves out from the source, the initial energy released in the seismic wave is spread over an increasing area and therefore the intensity of the wave decreases with distance (the case of geometric spreading).

Line 242. Low incident angles… which values? where?

Line 248. "highly attenuating". This is not modelled, so irrelevant (Applied)

Line 250. Those waves are not surface waves, but interface waves (Applied)

Line 251. I do not understand this argument. It seems wrong. In #6, the maximum amplitude is higher than in #5 indeed, but lower than in other scenarios without water. Can you explain (in the manuscript as well)?

(not sure I understand this point. the presence of water has an impact on radial strain, simply because part of the incident P-wave is converted to a tube wave (surface wave). This mean lower strain at the fibre. This conversion does not happen for scenarios 1-4. As a result, they (scenarios 1-4) have a larger response)

Figures 6-11 should display similar color scale and range so we can compare results.

(Unfortunately, we do not have access to the COMSOL anymore and our license has expired)

Fig. 12.15 could be shown all together for easier comparison. I do not get the benefice of comparing 2 by 2.

 (If they were all on the same graph, there would be too much information for readers to handle.)

Figure 7 to 11. "Radial". I really do not follow how the radial strain around the cable has this shape. Where is the wave coming from? Has the wave a waveform? The strain is changing with time, at which time is it represented? The scale between the different figure should be the same. The mesh seems better defined in Fig 8 and 9 compared to the figure 7-8 in the most central part. Why is this? Would it be possible to zoom closer to the cable to see the shape of the strain inside the cable? Figure 12. and 13. Indicate on the figure where is soft and hard.  Reporting scenario # is nice, but not very clear enough. The caption could get the list of scenarios main characteristics this comment is not mandatory, but it would make the reading much simplified). I see on figure much less than 10% difference between the curves, but at line 221, there is another value indicated, make it consistent.

The seismic pressure equation is used to construct prescribed displacement boundary conditions to ensure that the displacement is properly in a plane based on wave propagation. As a result, the wave does not have a specified waveform.

As the source operates at a dominant frequency, the simulation took place in the frequency domain (100 Hz). This type of simulation aids in the analysis of outcomes at the required frequency. As a result, there is no time limit.

Figure 14. line 285. Rephrase: "strains" are not "angles". (Applied)

I still do not understand why it is not possible to draw a figure with few loops of the 3d helix and show us how strain components vary along the fibre, from the 3D computation.

In this simulation, it is aimed to investigate the strain of cable in radial and axial directions and relate these strains to the strain of fiber, using equation A.49. So fiber strain for HWC is not a direct output from 3d modeling. However, radial and axial strains were the direct outputs from 3d models which were imported into Matlab to derive fiber strain.

Discussion.

As presented here this is not a discussion. A discussion would make the possible limitation of your study, and how to improve them. It would also be the location to discuss how to improve the deployment. Do you have suggestions, etc…

(Discussion was modified accordingly)

Line 292.  For completeness, I would also indicate Reinsch et al, 2017. Reinsch T., Thurley T., Jousset P., 2017. On the mechanical coupling of a fiber optic cable used for distributed acoustic/vibration sensing applications — a theoretical consideration. - Measurement Science and Technology, 28, 12. http://doi.org/10.1088/1361-6501/aa8ba4 (Applied)

Line 295. This is not shown at all. You never made the link between your theoretical study and the new Afton experiment that we hear about now only. In addition, this is not the place in the discussion. Why not starting the paper from this observation and justify all efforts. The introduction would be much easier to set up, and this would give an objective clear for your study. (Applied)

Line 301. Space missing between 10 m and for. (Applied)

Line 310. Was the straight cable a Constellation fibre? And what about the helically one?

(all fibre-optic cables results shown in this paper were obtained with Constellation fibre (this is clarified in the text in the section on New Afton data.)

Line 370 – figure 16. What is shown? Longitudinal strain? Radial strain? What is the color map used? )

( seismic data (ie strain rate) are shown in this figure. This is clarified in the text. The color bar description is added in the figure caption)

The reference list is to be checked. For example, Innanen et al. 2019 is not cited in (Fixed)

the manuscript.

Appendix A.

Line 441. You cannot start the appendix by "this". Which one? (Applied)

Line 443. The potential function letter (phi) is missing. (Applied)

Line 447. What is $\nabla$? (Applied)

Line 453. You mean A.1? (Applied)

Line 460. What is a "perfect" plane wave propagation?

(Because the source is far away, the spherical head wave is treated as a plane wave. As a result, displacement curve is in the plane)

Equations A5 and A6. Where is the shear stress component? Do you develop the method only for P-wave? This is very limiting… How would it be to expand to S-wave as well?

(For both analytical and numerical modeling: "S-wave could also be modeled for isotropic cases by considering that particle displacement is orthogonal to the wave propagation direction and adjusting the incidence angle accordingly)

Line 467-469. This approximation is really strong. A better quantification on the

error made should be properly given.

(This is simply describing a 2D simulation scenario which obviously has its limitations. This choice (2D modeling) has been explained in the section on numerical modeling. In Appendix A, we are providing the development for this 2D analytical response)

Line 470. You did not indicate what are the layers. Otherwise it is a bit more criptic…

Line 475-477. The shear stresses do exist! But where is $\sigma zz$? (Applied)

Line 477. What is this reference to "After 13"?? It seems you copied your text from

somewhere without checking properly the meaning of what you write… (Fixed)

Line 488. Again $\sigma zz$ is absent.

 (Because continuity equations between layers satisfied)

Line 490, why do you need the infinitesimal factor?

(Epsilon is used to satisfy the boundary condition within the range adjacent to the layers)

Line 491. Define d1, d2, etc… from fig A.2? (Applied)

Equation A42. not clear what F11, F12 etc… are. (Applied)

Line 506. A1 is the amplitude of the wave generated by the seismic source. It is known in the case of an active experiment. However, one may want to use helically fibre with microseismic data, where the source is unknown… Are we then lost?

(This research is intended to investigate the factors that influence HWC response, which necessitates the use of well-known parameters. The amplitude of the source can be determined for microseismic events by performing some back analysis)

The term $\varepsilon\, xz$ is missing, why? (Applied)

Line 513. This approximation is really the strong assumption of the approximation which allows you to migrate from 3D to 2D. This is exactly this difference we want to see between the two and validated by the 3D true numerical computation. As indicated by several referees, this equation is wrong sand should be corrected. As you see, there are still many unclarities, questions, and I guess my reading was not exhaustive. I would suggest you to read the manuscript again very carefully and bring all attention to make the manuscript readable, clearer and fully justified following the indications given by the referee reports and this review. Once those details will have been addressed, we can reconsider your manuscript.

(The formulation was not wrong, just notation was confusing, we changed the notation)

(The purpose of using a 2D analytical method is to validate the numerical simulations. In this simulation, it is aimed to investigate the strain of cable in radial and axial directions and relate these strains to the strain of fiber, using equation A.49 by assuming that the fibre is wrapped around the cable at an angle of 30 degrees. This wrapping angle is derived from the HWC cable used in the New Afton mine)

Referee 2

Minor comments:
Abstract: Typically an abstract does not contain paragraphs. please confirm with the journal's style guide. The first paragraph of the abstract reads more like an introduction. An abstract is supposed to be concise.

Authors action: The abstract has been shrunk in size to 232 words.

In section 2.2 you should clarify the source location (Source location is in the infinity, mentioned in the text) or how you get to retrieve the different incidence angles in Figures 4ff (The displacement boundary condition was established as a function of incidence angles ranging from 0 to 90 degrees) . Your workflow is not clearly described and thus not reproduceable (The formulas, boundary conditions (2d versus 3d), inputs, and other details were all given in detail). This is currently the major flaw of this manuscript (I hope this is no longer the case). What is your input amplitude of the wave? (1m) The model dimensions are missing (Size of the model for hard and soft formations are $4.78^3$ and $1.63^3$ m$^3$), as well as a description how values were extracted from the model to obtain the figures. It is good scientific practice to explicitly write the steps required to reproduce your work. A homogeneous (cement) block would be great as baseline model.

Your models and figures only show HWCs never straight cable. in the data examples in the end you then make conclusions about comparison. I may have missed some arguments here?
(The presenting sequence has been adjusted to make it more understandable to the readers).

Line 22: Two dots before "Results" (Applied)
Line 22: "modified" -> you never mentioned the unmodified model, so this seems out of place... (Applied)

Line 56: DAS can very well detect (dynamic) strain from hydro-fracs in vertical wells. It all depends on relative geometry and incidence angle (Applied)

Line 79: awkward grammar with 5 verbs in succession (Applied).

Line 96: be consistent in upper or lower case writing of "Appendix" (see line 102) (Applied)

Line 114: Dot missing after "processing" (Applied)

Line 130ff: This feels like a repetition from earlier. Maybe track-changes left-over fragment? (Applied)

Table 1: Scenario 5: What does "they are cable" mean???? (Deleted)

Table 2: your eta is physically identical to the wavelength, which is in seismology typically denoted with lambda (Eta was replaced by lambda)
Why did you chose 1/12 of the wave length? (Be able to Compare at a certain wave position)

Why are some parameters given in absolute diameters, others relative to wave length? (Be able to Compare at a certain wave position)
Which (dominant) frequency is assumed? (100 Hz)

Line 153: Dot missing at the end of sentence (Applied)

Figure 3: You never mention the (absolute) dimensions of you model (Size of the model for hard and soft formations are $4.78^3$ and $1.63^3$ m$^3$)

Line 170: Change to "Figure 4 and 5" (Applied)

Line 172: What units are the RMS errors (I suppose in "strains", typically denoted by epsilon)? Or is it in percent? (Unit is strain)

Figure 4: please clarify your angle definition. I suppose 90deg is cable parallel?(Based on $e_{\|(Fiber)} = e_{zz(Cable)} \cos^2 \alpha + e_{rr(Cable)} \sin^2 \alpha$, parallel to the cable means alfa 0)
is your "fibre strain" the ezz of Kuvshinov, or the strain in the surrounding material (ezz is the fibre strain).

Figure 5: Here are smaller max values than in Figure 4. Comment on this (Applied, Scenario 4 differs from the others in that the soft formation is adjacent to the casing in this case. Because of the huge difference in material properties, the reflection index on this surface is larger, and a large quantity of energy is reflected to the soft formation)
It may be instructive to have a simple model of a cable in a homogenous cement block as a baseline model....

Line 203: Fig 6 and 7 "compare" it is difficult to compare if the two have different color scales...
Line 207: Repeat WHY they cannot be compared (Applied)

Line 218: Check figure numbering, Fig 13 seems wrong here (Applied)

Line 223 "lesser" -> "smaller" (Applied)

Line 240f: The sentences starting with Scenario #5 /#6 can surely be simplified by mentioneing similarities and differences (text re-written with the following:  Figure 15 shows a comparison of modeled fiber strains for scenarios #5 and #6. These are both 5-layer scenarios similar to scenario #1, except for the presence of water either between the casing-formation annulus (scenario #5) or between the cable-casing annulus (scenario #6). Hard cement replaces water when not present for both scenarios.)

Fig 6-11: If I understand your paper correct you are modelling a plane wave traveling through your model. Here, you show a static strain "snap shot" of the wave? at what time step (It was in frequency domain)? How does that strain translate into DAS amplitudes (Amplitude is strain, isn't it?) How are these figures related to fig 4&5?

Line 241: This first paragraph of the discussion is in fact part of an introduction. I am sure the experience of the co-authors will help here to reformulate and restructure. (Applied)

CONCLUSION:
Explicitly write about benefits of HWC. At the moment it sounds very skeptical, but there seems to be features in the data that can only be explained when using HWCs (added one sentence highlighting the benefits of HWC at New Afton – mainly the identification of poorly cemented areas between the casing and rock formation)

---

## Editor Decision (ED2)

Dear Mostafa, dear authors,

Please apologies for too long process in managing your manuscript. This is indeed hard time to find referee willing reviewing. Therefore, as we need to progress and take a decision, and as I did not read the manuscript for some time, I reread it as if I was reading first time.

I am sorry to say, there is still one major issue which must be modified or clearly mentioned as a strong limitation in the approach, as the problem modeled is different, and this must be said. In short, the 2D approximation for a 3D helicoidal cable is simply wrong, or not justified enough. The issue is in the appendix A, where all the modeling development is performed. At line 507, you mention that "because the plane wave incident upon layer 1 is assumed to lie in the X-Z plane", then "the problem becomes 2D". This is simply wrong. I try to show it in the figure below.

[Figure]

On the left, the 3D real situation is represented, on the right the 2D approximation. In the 3D case, even if the incident plane wave is in the plane XZ, there will be rays (example R3) which will reflect in the layers with a different path as the one in the XZ plane, as they hit the layer one with a different incident angle, and therefore will modify amplitudes of the signal catched by both fibres, linear or helicoidal. In 2D, this is much more simple to evaluate indeed, as R3 will not interact with the plan XZ at the location of the cable, and will behave as R1 and R2, and not interact with R1 and r", contrary to the 3D situation.

Therefore, the approximation of the 3D into 2D is completely wrong and you cannot claim that the equations you derive are an approximation for the 3D case. As long as I do not see a true comparison between the real situation modeled in 3D and the 2D approximation, I would not agree that the discrepancy is small enough that your results represent an approximation of the 3D.

Now, indeed you can estimate what would be the amplitudes for a 2D case, but the problem addressed is completely different. Actually, even in this 2D approximation, as the fibre is helicoidal, it is really 3D, and therefore you cannot assume everything is happening in 2 dimensions.

Those point have been actually raised by all reviewers, but not properly addressed so far.

In addition, there are minor elements:

line 16 and line 19: you define incident angle and Aftor mine at the wrong place. Those terms are already used earlier in the abstract, spo move the details to the first instance

Line 46: The year of the reference Kuvshinov is not correct

Line 58 why only the latter is dependent on the material around?

Line 136: typo at analytical

Line 184: scenario 5: 2 times cable?

Figure 3. The text are much too small. Please increase the text size.

Line 236. a dot is missing at the end of the sentence.

Line 255. I do not understand the issue with the graphics.

Figure 11: Why the plot is not symetric? What is the reason ? Is there a specificity in this case?

Why do you show only cases where the incident angle is 90°?

Line 380. How more performant HWC systems could be better designed with an approximated modellig? It would be important to mention that a full 3d modeling is required.

References: Daley, 2016; Innanen et al., 2019, Reinsch et al., 2017  are not cited in the text

Line 447: please complete the reference, with all authors.

Line 482:  What is " $*$ " is the formulas A.1, A.8, A.9 etc… If simple multiplication, it should be removed, like the product between w and t is written wt and not w*t.

Line 485. Not clear what Nabla is. I guess is it not a collection, but a symbol for an operator. There is also 2 times "is".

Line 419. remove the upper case B at because.

Line 505. Please make clear what are the "multilayered media". I suppose those are the different concentric layers (cement, water, etc.) and not the geological layers.

Line 532. not clear "… and using = n, x= hn, …"

Best regards, Philippe Jousset, 17.06.2022

---

## Author Response (AR3)

Dear Editor,

First and foremost, thank you for your helpful feedback. We made a concerted attempt to respond to most of the comments clearly and concisely as follows:

***Editor's letter:***
I am sorry to say, there is still one major issue which must be modified or clearly mentioned as a strong limitation in the approach, as the problem modeled is different, and this must be said. In short, the 2D approximation for a 3D helicoidal cable is simply wrong, or not justified enough. The issue is in the appendix A, where all the modeling development is performed. At line 507, youmention that "because the plane wave incident upon layer 1 is assumed to lie in the X-Z plane",then "the problem becomes 2D". This is simply wrong. I try to show it in the figure below.
On the left, the 3D real situation is represented, on the right the 2D approximation. In the 3D case, even if the incident plane wave is in the plane XZ, there will be rays (example R3) which will reflect in the layers with a different path as the one in the XZ plane, as they hit the layer one with a different incident angle, and therefore will modify amplitudes of the signal catched by both fibres, linear or helicoidal. In 2D, this is much more simple to evaluate indeed, as R3 will not interact with the plan XZ at the location of the cable, and will behave as R1 and R2, and not interact with R1 and r", contrary to the 3D situation.
Therefore, the approximation of the 3D into 2D is completely wrong and you cannot claim that the equations you derive are an approximation for the 3D case. As long as I do not see a true comparison between the real situation modeled in 3D and the 2D approximation, I would not agree that the discrepancy is small enough that your results represent an approximation of the 3D. Now, indeed you can estimate what would be the amplitudes for a 2D case, but the problem addressed is completely different. Actually, even in this 2D approximation, as the fibre is helicoidal, it is really 3D, and therefore you cannot assume everything is happening in 2 dimensions. Those point have been actually raised by all reviewers, but not properly addressed so far.
Response: Thank you Philippe! I agree with you! We decided to focus more on 3-D modeling aspect of HWC. As a result, the analytical part and 2D model have been deleted from the paper. However, we would like to clarify that we didn't want to compare 3D with 2D model/analytical model, and 2D model was compared with analytical equations to assure that the boundary conditions have been set correct. It was explicitly stated that "However, results between 2D and 3D simulations cannot be directly compared quantitatively."

In addition, there are minor elements:

line 16 and line 19: you define incident angle and Aftor mine at the wrong place. Those terms are already used earlier in the abstract, spo move the details to the first instance
Response: Addressed

Line 46: The year of the reference Kuvshinov is not correct
Response: Addressed
Line 58 why only the latter is dependent on the material around?
Response: Addressed, we meant more dependent

Line 136: typo at analytical
Response: Deleted

Line 184: scenario 5: 2 times cable?
Response: Deleted

Figure 3. The text are much too small. Please increase the text size.
Response: Addressed

Line 236. a dot is missing at the end of the sentence.
Response: Addressed

Line 255. I do not understand the issue with the graphics.
Response: The picture below should clarify that there are no meshing issues with COMSOL. Inside the domain, a tetrahedral mesh was employed, while lower reflecting surfaces were covered with a swept mesh.

[Figure]

- *This Type of Mesh is mapped mesh. It is strongly recommended to use mapped meshing in the infinite boundaries to prevent poor mesh element quality, giving rise to poor or slow convergence for iterative solvers and making the problem ill-conditioned in general.*

*Plan View*

*Side View*

*Front View*

Figure 11: Why the plot is not symetric? What is the reason ? Is there a specificity in this case?
Response: The seismic load is applied on one side in this simulation (the right-hand side of the model). Because of this, the strain distribution around the borehole shouldn't generally have a symmetrical pattern (as you can see even in the other scenarios even slightly). The reflection coefficient is higher in this scenario than in the others due to the notable change in material properties between the cable and the soft cement (formation), which makes the asymmetric pattern more obvious.

Why do you show only cases where the incident angle is 90°?
Response: Maximum fiber strain occurs at 90°. On the graphs 13 to 16, the results for lower incident angles are given.

Line 380. How more performant HWC systems could be better designed with approximated modeling? It would be important to mention that full 3d modeling is required.
Response: The HWC/straight fibre installation in the field could be guided by this 3D model, and the results could even be interpreted using it. For instance, as shown in scenario 6, the fibre will be under extension rather than compression at an angle between 10 and 30. The fibre won't be able to record the strain if it occurs in the field, and the numerical model can predict/justify it. These findings may also indicate the conditions in which fibre will perform the best.

References: Daley, 2016; Innanen et al., 2019, Reinsch et al., 2017 are not cited in the text
Response: addressed

Line 447: please complete the reference, with all authors.
Response: Addressed

Line 482: What is " " is the formulas A.1, A.8, A.9 etc... If simple $*$ multiplication, it should be removed, like the product between w and t is written wt and not w*t.
Line 485. Not clear what Nabla is. I guess is it not a collection, but a symbol for an operator. There is also 2 times "is".
Line 419. remove the upper case B at because.
Line 505. Please make clear what are the "multilayered media". I suppose those are the different concentric layers (cement, water, etc.) and not the geological layers.
Line 532. not clear "... and using = n, x= hn, ..."
Best regards, Philippe Jousset, 17.06.2022

Response: Appendices are deleted.

---

## Author Response (AR4)

Dear Editor,

First and foremost, I want to thank you for your insightful feedback, which has improved the quality of our paper. We made a conscious effort to address the comments concisely and clearly as follows:

line 28. The reference to Madjdabadi, 2016 need to be adapted to the real name. Is it Mollahasani (as indicated in line 372?). please adjust one or the other.
Adjusted

Line 36. reference Daley et al., 2016 misses "et al."
Adjusted

Line 46. reference Mateeva et al., 2014, should not be italic.
Adjusted

Line 46. Kuvshinov 2016 reference is missing in the reference list.
Added

Line 54. For completeness you could add a reference about this ability to measure complex wavefield. Suggestion: Jousset, P., Currenti, G., Schwarz, B. et al. Fibre optic distributed acoustic sensing of volcanic events. Nat Commun 13, 1753 (2022). https://doi.org/10.1038
Adjusted

Line 77. This sentence has no verb...
Added

Line 158-159. This setence is a repetition of the sentence at lines 148. You could consider group them.
Grouped together

table 2. I miss this before... what is 12?
Our intention was to have the model dimension as a function of wavelength to compare all the scenarios at specific location of wave.

Line 201. There are 2 times a dot '.'. One is enough ;-)
Deleted.

Line 245. I guess this is the most important result. You could mention what is the main reason for the attenuated response of the HWC instead.
Applied in the text.

Line 302. remove the space between "on" and "e" to make the proper word "one".

Deleted

Line 334. I did not find a call to the reference Brinkgreve in the text. Either introduce a reference call in the text or remove it.
Deleted

Thank you for considering our manuscript for publication in Solid Earth.

Sincerely,
Mostafa Gorjian, On behalf of all the authors